# ARTICLES

# Reference genome assemblies reveal the origin and evolution of allohexaploid oat

Yuanying Peng [1,2,3,14,15 ✉], Honghai Yan[1,3,14], Laichun Guo[2,4,14], Cao Deng[5,6,14], Chunlong Wang[2,4,14], Yubo Wang[7,14], Lipeng Kang [8,9], Pingping Zhou[1,3], Kaiquan Yu[3], Xiaolong Dong[3], Xiaomeng Liu[3], Zongyi Sun[10], Yun Peng[3], Jun Zhao[3], Di Deng[3], Yinghong Xu[3], Ying Li[3], Qiantao Jiang [1,3], Yan Li[1], Liming Wei[2,4], Jirui Wang[1,3], Jian Ma [1,3], Ming Hao [1,3], Wei Li[1,3], Houyang Kang[1,3], Zhengsong Peng[11], Dengcai Liu[1,3], Jizeng Jia [12], Youliang Zheng[1,3], Tao Ma [7,15 ✉], Yuming Wei [1,3,15 ✉], Fei Lu [8,9,13,15 ✉] and Changzhong Ren [2,4,15 ✉]

**Common oat (*Avena sativa*) is an important cereal crop serving as a valuable source of forage and human food. Although reference genomes of many important crops have been generated, such work in oat has lagged behind, primarily owing to its large, repeat-rich polyploid genome. Here, using Oxford Nanopore ultralong sequencing and Hi-C technologies, we have generated a reference-quality genome assembly of hulless common oat, comprising 21 pseudomolecules with a total length of 10.76 Gb and contig N50 of 75.27 Mb. We also produced genome assemblies for diploid and tetraploid *Avena* ancestors, which enabled the identification of oat subgenomes and provided insights into oat chromosomal evolution. The origin of hexaploid oat is inferred from whole-genome sequencing, chloroplast genomes and transcriptome assemblies of different *Avena* species. These findings and the high-quality reference genomes presented here will facilitate the full use of crop genetic resources to accelerate oat improvement.**

Common oat (*A. sativa* L., 2n = 6x = 42, AACCDD genomes) is an important cereal crop cultivated worldwide and has long been prized by consumers primarily because it is one of the richest sources of protein, fat and vitamin B1 among all crops[1]. In addition, oats are a widely grown cool-season annual forage species, and represent a major source of high-quality forage for livestock globally[2]. Common oat belongs to the genus *Avena* in the grass family Poaceae, and the genus comprises a polyploid series of wild, weedy and cultivated species distributed across six continents[3]. The diploids have either the AA or CC genomes; the tetraploids mainly contain the AABB or CCDD[4] (previously AACC) genomes; all hexaploid species, including common oat, share the same AACCDD genomic constitutions[5]. Due to the lack of genome sequences from hexaploid oat and its close diploid and tetraploid relatives, the phylogenetic history and divergence time among the A, C and D genomic lineages remain unclear. Polyploid plants often have notable advantages in biomass production, vigor and adaptability to environmental changes, contributing to the emergence of important agronomic traits in food crops[6–8]. Oat has good adaptability to a wide range of climatic conditions, enabling oat to reliably produce grains in marginal regions with harsh conditions. Therefore, crop polyploidization plays an important role in next-generation crop improvement to overcome food security challenges[8]. Many genomes of commercially important crops have been sequenced and assembled, which has improved the understanding of crop evolutionary history and the development of efficient approaches for selecting important traits[9]. Oat has lagged behind in this regard, primarily due to the large genome[10] that contains highly repetitive DNA sequences, and the fact that both the A- and D- subgenomes are similar to the A-genome diploid and were difficult to distinguish from one another in previous studies[11]. Currently, relatively little is known regarding the position and distribution of genes on each of the oat chromosomes and their evolution during the polyploidization events that gave rise to the hexaploid species, which limits the full and effective utilization of oat germplasm. With the recent advances made by Oxford Nanopore Technologies (ONT), the ONT system now offers ultralong sequence reads, delivering high contiguity with low assembly errors caused by long repetitive regions[12]. This technology has facilitated complete telomere-to-telomere (T2T) genome assembly in various species, including *Homo sapiens*,

[1]State Key Laboratory of Crop Gene Exploration and Utilization in Southwest China, Sichuan Agricultural University, Chengdu, China. [2]National Oat Improvement Center, Baicheng Academy of Agricultural Sciences, Baicheng, China. [3]Triticeae Research Institute, Sichuan Agricultural University, Chengdu, China. [4]China Oat and Buckwheat Research Center, Baicheng, China. [5]The Key Laboratory of Animal Disease and Human Health of Sichuan Province, College of Veterinary Medicine, Sichuan Agricultural University, Chengdu, China. [6]Departments of Bioinformatics, DNA Stories Bioinformatics Center, Chengdu, China. [7]Key Laboratory of Bio-Resource and Eco-Environment of Ministry of Education, College of Life Sciences, State Key Laboratory of Hydraulics and Mountain River Engineering, Sichuan University, Chengdu, China. [8]State Key Laboratory of Plant Cell and Chromosome Engineering, Institute of Genetics and Developmental Biology, Innovative Academy of Seed Design, Chinese Academy of Sciences, Beijing, China. [9]University of Chinese Academy of Sciences, Beijing, China. [10]Grandomics Biosciences, Wuhan, China. [11]Panxi Crops Research and Utilization Key Laboratory of Sichuan Province, Xichang University, Xichang, China. [12]Institute of Crop Sciences, Chinese Academy of Agricultural Sciences, Beijing, China. [13]CAS-JIC Centre of Excellence for Plant and Microbial Science (CEPAMS), Institute of Genetics and Developmental Biology, Chinese Academy of Sciences, Beijing, China. [14]These authors contributed equally: Yuanying Peng, Honghai Yan, Laichun Guo, Cao Deng, Chunlong Wang, Yubo Wang. [15]These authors jointly supervised this work: Changzhong Ren, Fei Lu, Yuming Wei, Tao Ma, Yuanying Peng. ✉e-mail: yy.peng@hotmail.com; matao.yz@gmail.com; ymwei@sicau.edu.cn; flu@genetics.ac.cn; renchangzhong@163.com

**Table 1 | Genome assembly statistics for the diploid, tetraploid and hexaploid *Avena* species**

| | *A. longiglumis*, CN 58138 (2n = 2x = 14), AlAl | *A. insularis*, CN 108634 (2n = 4x = 28), CCDD | *A. sativa* ssp. *nuda* cv. Sanfensan (2n = 6x = 42), AACCDD |
|---|---|---|---|
| Illumina (Gb) | 204.68 | 451.89 | 649.68 |
| ONT (Gb) | 268.74 | 481.39 | – |
| ONT ultralong (Gb) | – | – | 1260.30 |
| Hi-C (Gb) | – | 816.93 | 1312.83 |
| IsoSeq (Gb) | 25.74 | 49.94 | 81.14 |
| Total assembly size (bp) | 3,736,548,545 | 7,519,018,440 | 10,757,433,345 |
| Longest contigs (bp) | 29,014,927 | 30,586,412 | 313,872,778 |
| Number of contigs | 956 | 2,732 | 436 |
| N50 contig length (bp) | 7,297,603 | 5,637,473 | 75,273,016 |
| L50 contig count | 160 | 399 | 44 |
| N90 contig length (bp) | 2,123,884 | 1,477,395 | 16,452,443 |
| L90 contig count | 523 | 1,374 | 160 |
| Number of contigs per (sub)genome | A: 943 | – | A: 87 |
| | – | C: 1,019 | C: 159 |
| | – | D: 848 | D: 77 |
| | Total: 943 | T: 1,867 | T: 323 |
| Sequences assigned per (sub) genome (bp) | A: 3,708,832,268 | – | A: 3,346,241,509 |
| | – | C: 4,020,068,809 | C: 4,094,063,504 |
| | – | D: 3,131,824,267 | D: 3,215,854,646 |
| | T: 3,708,832,268 | T: 7,151,893,076 | T: 10,656,159,659 |
| Genome completeness (BUSCO) | 98.51 | 99.32 | 99.44 |
| Repeats (%) | 86.83% | 87.11% | 86.95% |
| Protein-coding genes | A: 43,477 | – | A: 41,433 |
| | – | C: 43,243 | C: 36,283 |
| | – | D: 43,911 | D: 41,633 |
| | T: 43,477 | T: 89,995 | T: 120,769 |
| Pseudogenes | A: 14,058 | – | A: 23695 |
| | – | C: 20,847 | C: 28056 |
| | – | D: 17,766 | D: 24,009 |
| | T: 14,058 | T: 40,027 | T: 76,646 |

A, A subgenome; C, C subgenome; D, D subgenome; T, total.

by resolving long, complex repetitive regions[13,14]. It is very suitable for assembling the large, complex polyploid oat genome, with a high content of repetitive sequences and high subgenomic homology. Here, we used the ultralong ONT system to sequence the genome of the oat variety 'Sanfensan' (*A. sativa* ssp. *nuda*), a hexaploid oat landrace that originated from the diversity center of hulless oat, together with assembling and resequencing the genomes of additional diploid and tetraploid *Avena* species to gain insights into the evolutionary processes that established the dominant subgenome in allopolyploid oat species and demonstrate the utility of the oat reference genome in identifying genes that underlie agronomic traits.

## Results

**Assembly and annotation of the oat genome.** We assembled the Sanfensan genome into 326 contigs based on 1,028 Gb of ultralong reads (N50 length: 52 kb; ~100× genome coverage). The draft genome assemblies were then corrected by using ~650 Gb of cleaned Illumina paired-end reads, resulting in a total assembly size of 10.76 Gb, 99.06% of which was arranged into 21 chromosomes

(based on 1,296 Gb of Hi-C data) after dissociating 72 misjoined contigs into 182 contigs using three-dimensional proximity information (Extended Data Fig. 1a). The final assembly contained 436 corrected contigs with N50 of 75.27 Mb and a maximum length of 313.87 Mb. Of these, 323 contigs were anchored onto 21 pseudochromosomes (Table 1, Supplementary Tables 1–3 and Extended Data Fig. 1a). The quality of the Sanfensan genome assembly was supported by assessments including NG plots, long terminal repeat (LTR) assembly index (LAI; 18.34), BUSCO (99.44%) and the coverage of full-length transcripts by the assembled genome (92.00%) (Extended Data Fig. 2), together with the markers from hexaploid oat consensus map[15] and high level of syntenic relationship with the other hexaploid OT3098 v2 reference genome (https://wheat.pw.usda.gov/jb?data=/ggds/oat-ot3098v2-pepsico) (Extended Data Fig. 3). In addition, we found that 99.87% and 96.30% of the assemblies were covered at more than 20× sequencing depth with ultralong reads and short reads, respectively, indicating high accuracy at the nucleotide level (Extended Data Fig. 2d,e). Finally, 99.75% of the 4,336,693,678 Illumina paired-end reads could be mapped onto the assembly, with 98.22% properly paired alignments. From this

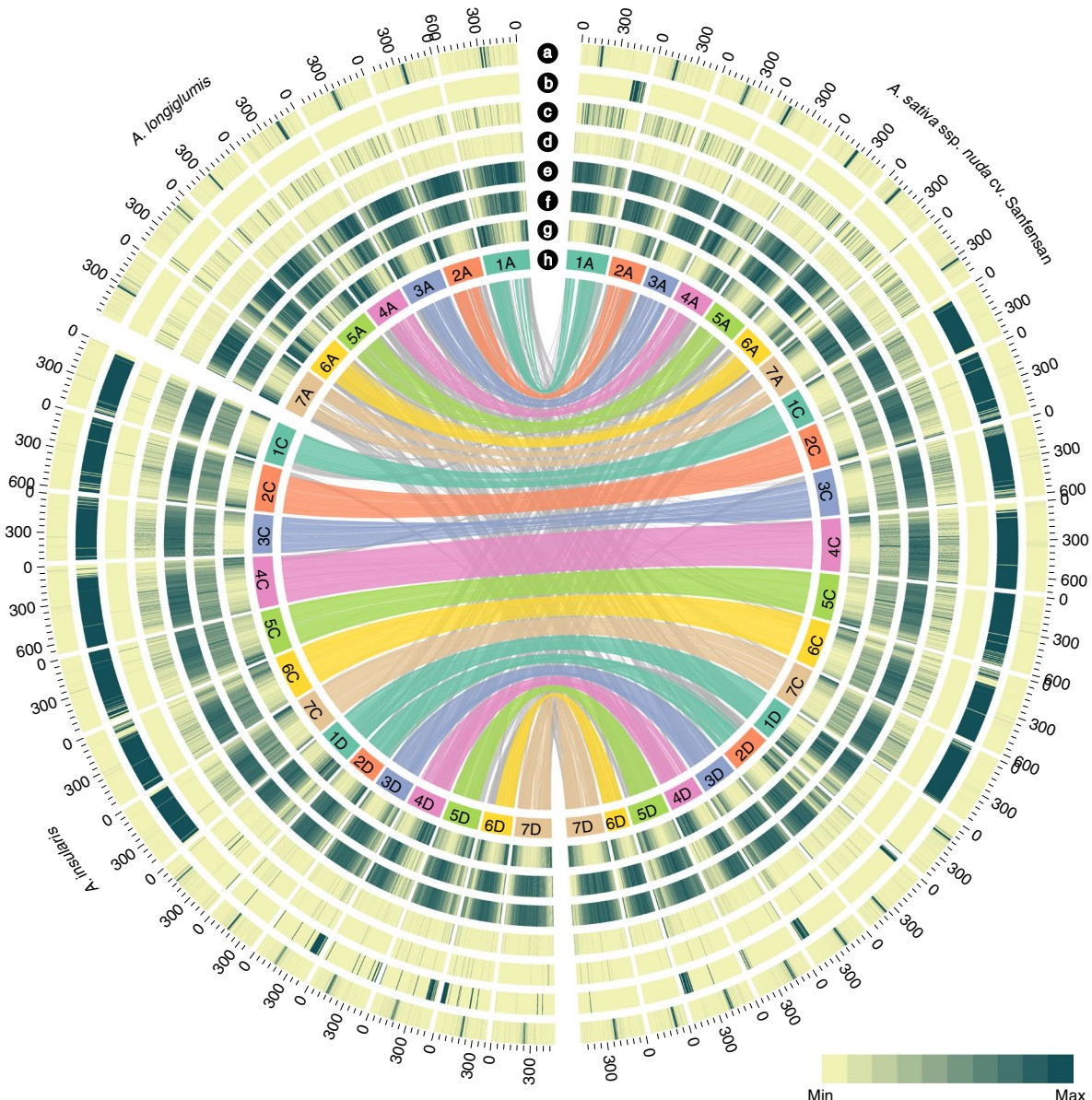

**Fig. 1 | Circos display of the genomic features of the assembled diploid *A. longiglumis*, tetraploid *A. insularis* and hexaploid *A. sativa* ssp. *nuda* genomes. a**, The inferred centromere positions of the A and D genome chromosomes. **b**, The distribution of the C genome-specific repeat Am1 along each chromosome. The Am1-rich regions on chromosomes 1A, 2D, 3D, 4D and 5D of Sanfensan are C genome introgressions. **c**, The distribution of the A genome-specific repeat As120a along each chromosome. **d**, k-mer frequencies. **e**, Tandem repeat (TR) density (MaxPeriod ≤500 bp). **f**, LTR retrotransposon density. **g**, Gene density. **h**, Chromosome names and sizes. The innermost layer shows synteny of hexaploid and its ancestor species, with colored upper-layer links representing syntenic blocks of each hexaploid chromosome and its ancestor chromosome, and the gray low-layer shows chromosome rearrangements during hexaploidization.

mapping result, 98,885 homozygous single-nucleotide polymorphisms (SNPs) and 19,444 insertions/deletions (InDels) were identified in Sanfensan, giving an estimated overall nucleotide accuracy rate of 99.999%. We also identified 1,093,198 heterozygous SNPs and 66,019 heterozygous InDels, giving an overall heterozygosity rate of 0.011%, indicating that the Sanfensan genome is largely homozygous. We annotated a total of 120,769 protein-coding genes, 88.41% of which were assigned to a predicted function and 33.69% of which have alternative splicing (AS) variants (Table 1, Supplementary Tables 4 and 5). Moreover, 86.95% (9.35 Gb) of the assembly was annotated as repetitive elements (Supplementary Table 6), which is higher than previously reported genomes of barley (80.80%)[16] and bread wheat (84.70%)[9].

To distinguish the subgenomes accurately and clarify the polyploidization history of the hexaploid oat, we sequenced and assembled its most likely ancestral species *A. longiglumis* (2n = 2x = 14, AlAl genome) and *A. insularis* (2n = 4x = 28, CCDD genome)[5], resulting in >60× genome coverage for *A. longiglumis* (218.67 Gb) and *A. insularis* (374.77 Gb). The assembled *A. longiglumis* genome was 3.74 Gb in size, 99.20% of which was anchored onto seven chromosomes with the *A. atlantica* genome as the reference[17]. The tetraploid *A. insularis* genome assembly was 7.52 Gb in size, 95.08% of which was arranged into 14 chromosomes by Hi-C analysis (Extended Data Fig. 1b). We annotated 43,477 and 89,995 protein-coding genes in the *A. longiglumis* and *A. insularis* genomes respectively (Table 1). Similarly, most of the

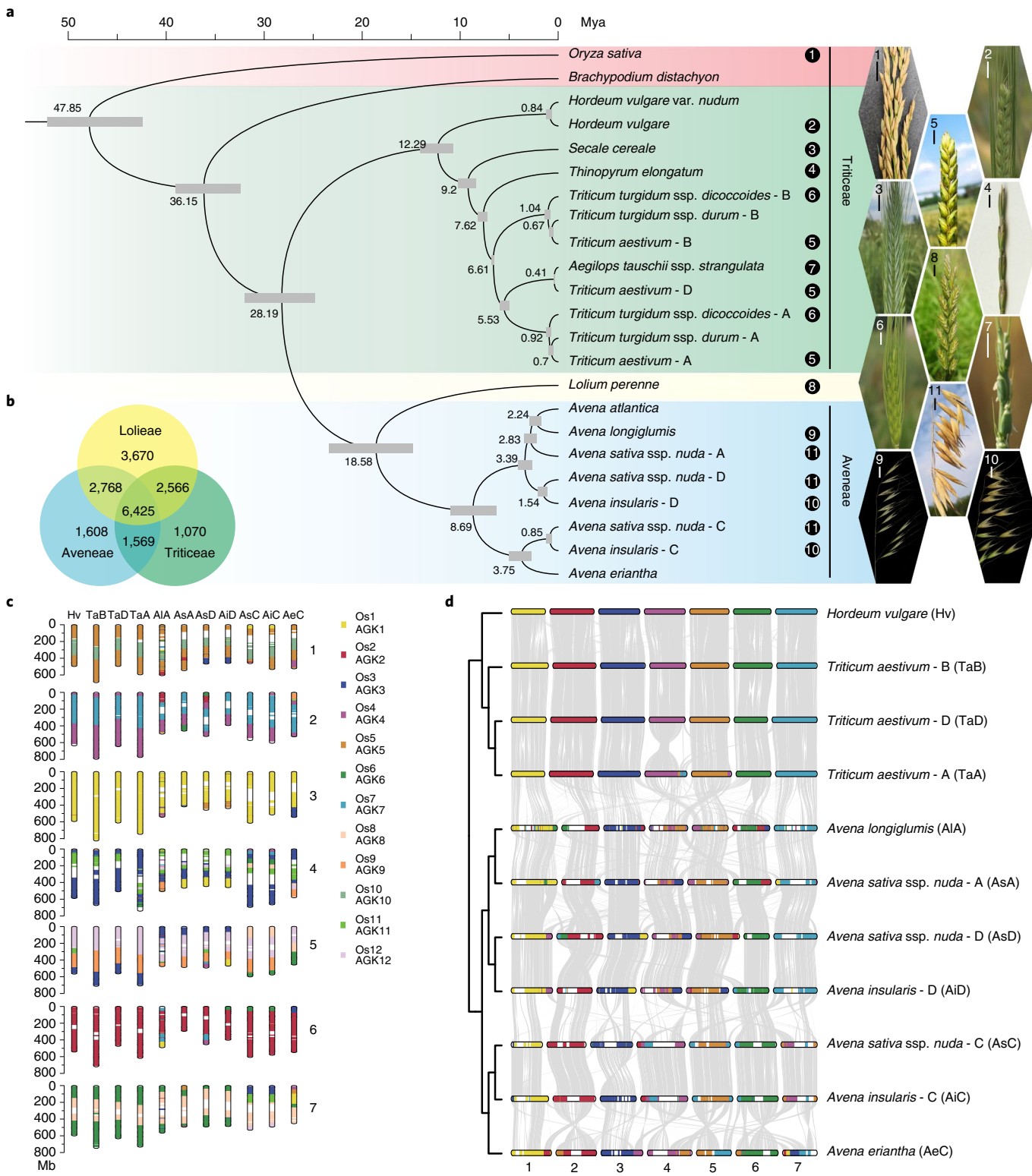

**Fig. 2 | Phylogenomic relationships of cereal crops. a**, Phylogeny and time scale of 23 subgenomes from 16 cereal crops and related grass species, including panicle images of the 11 sampled cereal crops. The number in the upper left corner of each panicle image corresponds to the number following the species name in the phylogenetic tree. Scale bars, 1 cm. **b**, Venn diagram showing the numbers of shared and unique gene families in the tribes of Aveneae, Lolieae and Triticeae. **c**, Probable chromosome evolutionary scenario of oat and wheat species. The subgenome chromosomes (1–7) are presented with a color code to show different segments from the 12 chromosomes of rice (Os1–Os12), which can be used as the representative of the ancestral grass chromosomes (AGK1–AGK12). The abbreviations in the top of the subgenomes are accordance with the abbreviations indicated in the brackets of panel d. **d**, Chromosome evolution based on barely genomes and chromosomal synteny between the three subgenomes of oat and wheat.

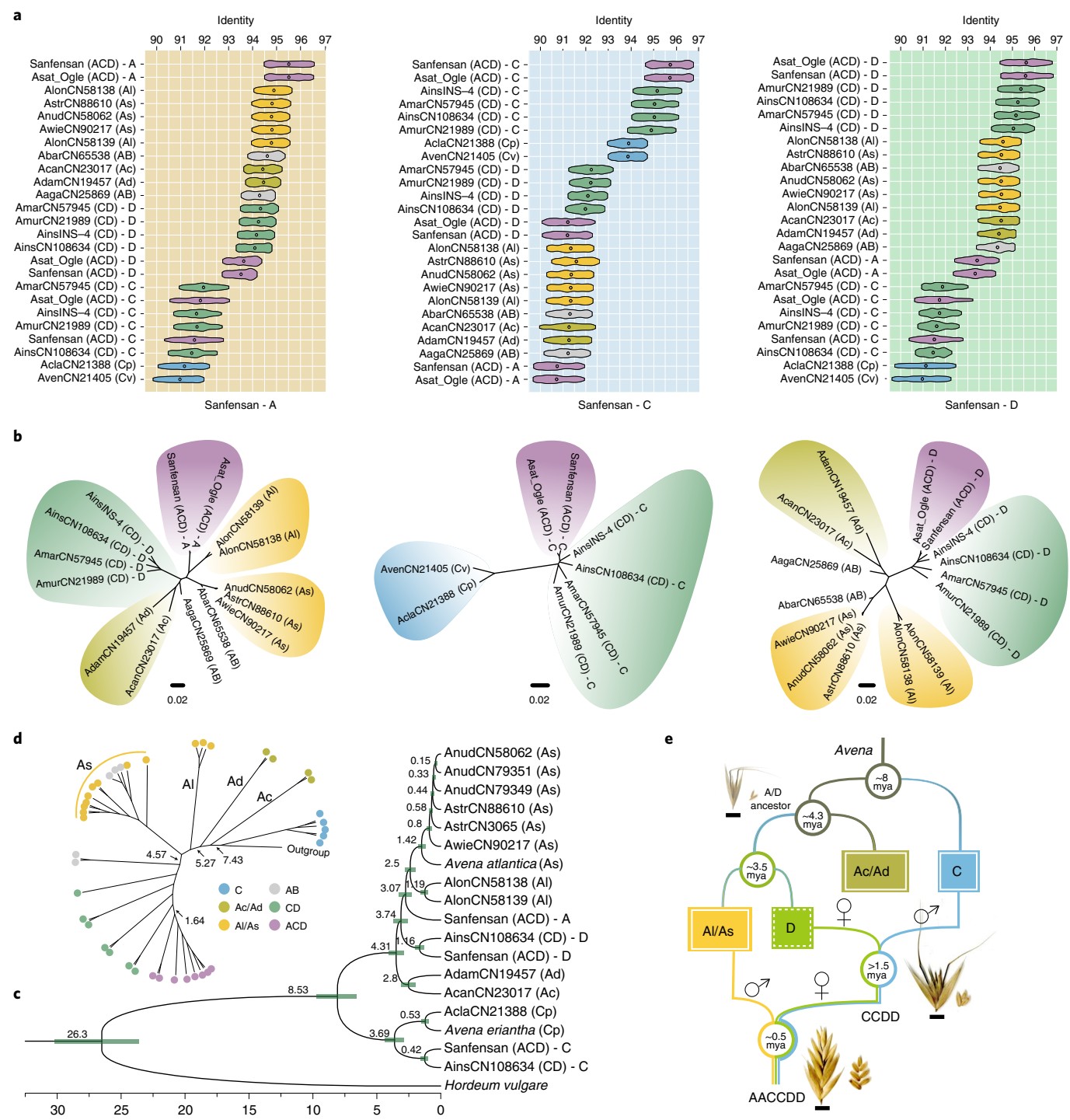

**Fig. 3 | Phylogenetics and polyploidization of *Avena* species. a**, Sequence similarities of reads from different *Avena* species that were uniquely mapped to the A, C and D subgenomes of Sanfensan. **b**, Phylogenetics of the A-, C- and D-genome lineages generated from the SNPs. **c**, Phylogenetic analysis based on 7,353 nuclear genes using barley as the outgroup. **d**, Unrooted phylogenetic tree constructed using the complete sequences of the *Avena* chloroplast genomes. **e**, Model showing the evolutionary history of hexaploid oat (*A. sativa* ssp. *nuda*; AACCDD).

*A. longiglumis* (86.83%) and *A. insularis* (87.11%) genomes are composed of repetitive elements (Table 1 and Supplementary Table 6).

Based on the genomic similarity among these three genomes, we successfully assigned the 21 pseudochromosomes of hexaploid oat to the A, C and D subgenomes (Extended Data Fig. 4a–d). The subgenome assignment was supported by the over-representation of the A- and C-genome-specific satellite repeats, As120a and Am1, respectively[18,19], in the corresponding subgenomes (Fig. 1), and by mapping the whole-genome sequencing reads from a range of *Avena* species (Extended Data Fig. 4e,f). The nomenclature system of OT3098 v2 was adopted for naming the chromosomes of Sanfensan, which was consistent with that approved by the International Oat Nomenclature Committee.

**Evolutionary position of oat among cereal crops.** The Poaceae family consists of many agronomically important species, commonly known as cereals, that are classified into three subfamilies: Oryzoideae (rice), Panicoideae (maize, sorghum) and Pooideae (Triticeae: wheat, barley and rye; Aveneae: oat). Among them, Oryzoideae and Pooideae belong to the BOP clad (Bambusoideae, Oryzoideae and Pooideae), which is one of the two primary groups in the Poaceae family. To further clarify the evolutionary position of oat among cereal crops, we used our oat reference genome assemblies to perform phylogenomic analysis to include oat. Gene family analysis across 23 subgenomes from 16 species belonging to the BOP clade identified 2,237 single-copy orthologs (Supplementary Tables 7–9). Phylogenomic analyses (Fig. 2a) revealed that the divergence between Aveneae and Triticeae took place after the speciation of Oryzoideae, with the approximate times for the two events being 28.2 and 47.9 million years ago (mya), respectively. The tribes Aveneae and Lolieae were indicated to be more closely related than Triticeae. The diversification of oat species occurred ~8.7 mya, which is earlier than wheat species (~6.6 mya) and falls within the previously estimated speciation time of *Avena* diploids (5.4–12.9 mya)[17]. We compared the gene families among Aveneae, Triticeae and Lolieae and found 6,425 common gene families in these three tribes, and there are 1,608 gene families specific to Aveneae (Fig. 2b). The top three most enriched Gene Ontology (GO) terms are 'response to auxin', 'enzyme inhibitor activity' and 'transcription factor activity' (Supplementary Table 10).

The ancestral grass karyotype (AGK) was previously reconstructed as a post-ρ AGK with 12 protochromosomes and a pre-ρ AGK with seven protochromosomes by comparing extant species[20]. Rice was identified as the most slowly evolving species and has 12 chromosomes, most of which closely resemble the post-ρ AGK. Based on the phylogenomic analysis, Triticeae (wheat, barley and rye) and Aveneae (oat) clustered together with Oryzoideae (rice) as an outgroup. We therefore used the rice genome as the ancestral reference and the barley genome as a closely related reference to investigate the chromosomal evolution of oat and to compare it with another allohexaploid cereal species, wheat. We identified 733 and 872 syntenic blocks that included 33,065 and 37,042 orthologous gene pairs from the subgenomes of hexaploid oat based on rice and barely, respectively. (Fig. 2c and Supplementary Table 11). Essentially, the third homologous group of oat and wheat was derived from a single ancient chromosome, AGK1 or Os1, except for the AGK3 or Os3 segment that was translocated to chromosome 3C in *A. eriantha*. The second group was mainly derived from the insertion of AGK7 or Os7 into AGK4 or Os4, and the remaining five groups were each derived from at least three ancestral

chromosomes via complex translocations (Fig. 2c). When the oat assembly was compared with the three subgenomes of common wheat using barley genomes as a reference, a large number of chromosomal rearrangements were identified. For example, although the second homoeologous group of oat and wheat are mainly derived from AGK4 or Os4 and AGK7 or Os7, the arrangement patterns of these two ancestral chromosomes are different.

**Polyploidization history and reticulate evolution.** Previous phylogenetic studies could not be well distinguish between the A and D subgenomes of *Avena*, which led to confusion and gave inconsistent results about the origins of hexaploid oat[3–5,11,21]. We have now definitely assigned chromosomes to the A and D subgenomes with the resolution afforded by the complete genome assemblies. Based on this foundation, and to identify the subgenome donors accurately, we resequenced the genomes of 14 *Avena* species representing different genomic subtypes and ploidy levels (As, Al, Ac, Ad, Cv, Cp, AB and CD). A total of 5,014.96 Gb of genome resequencing data (Supplementary Table 1) were mapped against the A, C and D subgenomes (Methods and Supplementary Table 12). The identity distribution clearly showed that the Al/As diploid species, and the C and D subgenomes of *A. insularis* have the highest similarities with the A, C and D subgenomes of hexaploid oat, respectively (Fig. 3a). The SNP-based phylogenetic tree of the three subgenomes consistently implies that the species cluster according to the genome subtypes, in which Al and As are the most closely related to the A subgenome of hexaploid oat, whereas none of the extant diploids shows a particularly close relationship with the C and D subgenomes (Fig. 3b). These results are further supported by the phylogenetic analyses of transcriptome data from 11 diploid species (Fig. 3c), suggesting that the A-genome progenitor of hexaploid oat was more likely the ancestor of the Al and As diploids than an extant diploid species, which may be the reason that previous studies[22] have considered different A diploids to be the A-genome donors of hexaploid oat.

We also assembled the chloroplast genomes of different *Avena* species (Supplementary Table 13), as the maternal inheritance characteristic can be used to determine the maternal genome donor to hexaploid oat. By including 25 previously reported chloroplast genomes (Supplementary Table 14)[22,23], our phylogenetic analysis (Fig. 3d) showed that the C genome was undoubtedly the male parent in the polyploidization and that the D genome, rather than the A genome, was the maternal donor in hexaploid oat. The evolutionary order of the different A-genome subtypes was Ac-Ad-Al-As (Fig. 3d). The relatively low collinearity between the C genomes of the diploid and polyploid species is consistent with the

**Fig. 4 | Subgenome evolution in *Avena* species. a**, Synteny between the subgenomes of hexaploid oat and the putative tetraploid and diploid ancestors. The yellow and blue arrows and lines connecting the chromosomes represent the observed large chromosomal translocations (>40 Mb) from the A and C genomes, respectively. The dark gray arrow and line indicate the inversion in chromosome 3C between *A. insularis* and Sanfensan. **b**, Large C-to-A or C-to-D translocations are supported by mapping reads from the C genome diploid to the hexaploid reference genome; blue arrows indicate C-to-D and C-to-A intergenomic translocations. **c**, FISH using the C genome-specific repeat as a probe confirms the C-to-A and C-to-D translocations. Fluorescence signals from the A genome-specific repeat (As120a) are shown in green, and signals from the C genome-specific repeat (Am1) are in red. The white arrows indicate C-to-D and C-to-A intergenomic translocations. **d**, FISH confirms the inversion on chromosome 3C observed in the hexaploid oat genome. Oligo-5SrDNA (red) and Oligo-6C343 (green) gave clear hybridization signals on the short and long arms of the tetraploid 3C, respectively, whereas both signals are observed on the long arm of chromosome 3C in the hexaploid. For karyotyping in panels c and d, at least three slides for each accession and ten chromosomes per slide were examined. Scale bars, 2 μm. **e**, Comparison of *Ka/Ks* value distributions between the three subgenomes of hexaploid oat. The central line for each box plot indicates the median. The top and bottom edges of the box indicate the first and third quartiles and the whiskers extend 1.5 times the interquartile range beyond the edges of the box. Numbers of samples used in each assay are indicated as *n*. The asterisks represent significant differences (two-tailed Wilcoxon rank-sum test, \*\*$P < 0.01$). **f**, Two-dimensional hierarchical cluster analysis of gene expression among single-copy homoeologous oat genes compared with organ-specific gene expression. TPM, transcripts per million. **g**, Analysis of $\log_2$-fold changes in pairwise gene expression between homoeologous genes showed biased expressions. DEGs, differentially expressed genes. Dot plots show the fold changes for each triplet ordered as shown in the y axis (f). The numbers of significantly differentially expressed triplets (one-tailed Fisher's exact test, $P < 0.05$) across all organs are shown at the top of the box, and the significance of the differences in the values between subgenomes was estimated using the one-tailed Wilcoxon rank-sum test (\*$P < 0.05$, \*\*$P < 0.01$).

nuclear–cytoplasmic interaction hypothesis, which suggests that the paternally inherited genome of an allopolyploid is usually more prone to genetic changes than the maternally derived genome[24].

In addition, our results indicate that the D-genome progenitor of hexaploid oat is more closely related to the A-genome than to the C-genome and may be extinct. The C and A/D lineages diverged approximately 8 mya, followed by the A-genome subtypes (Ac/Ad) and the D genome around 3.5 mya. Cultivated ACD-genome hexaploid oat originated around 0.5 mya from the hybridization between a paternal Al/As-genome diploid ancestor and a maternal CD-genome tetraploid that is closely related to *A. insularis* and originated from an allotetraploidy event between a paternal C-genome and a maternal D-genome diploid (Fig. 3e). These findings clarified

the evolutionary history of oats based on various pieces of evidence at the genomic level and provided the most possible clues to the subgenomic origin of hexaploid oat, which will be of great value for introgression breeding and the transfer of traits from the closest extant wild relatives (As/Al genome diploids and CD genome tetraploids) to cultivated oat.

**Subgenome structure and dominance.** Consistent with the inference that the A and D subgenomes of oats are closely related, the synteny between them (54.40% genes in collinear blocks) is more extensive ($P < 2.2 \times 10^{-16}$, Fisher's exact test) than that between the C and A/D subgenomes in hexaploid (43.59% in C and A and 41.43% in C and D) and tetraploid (38.81% in C and D) (Extended

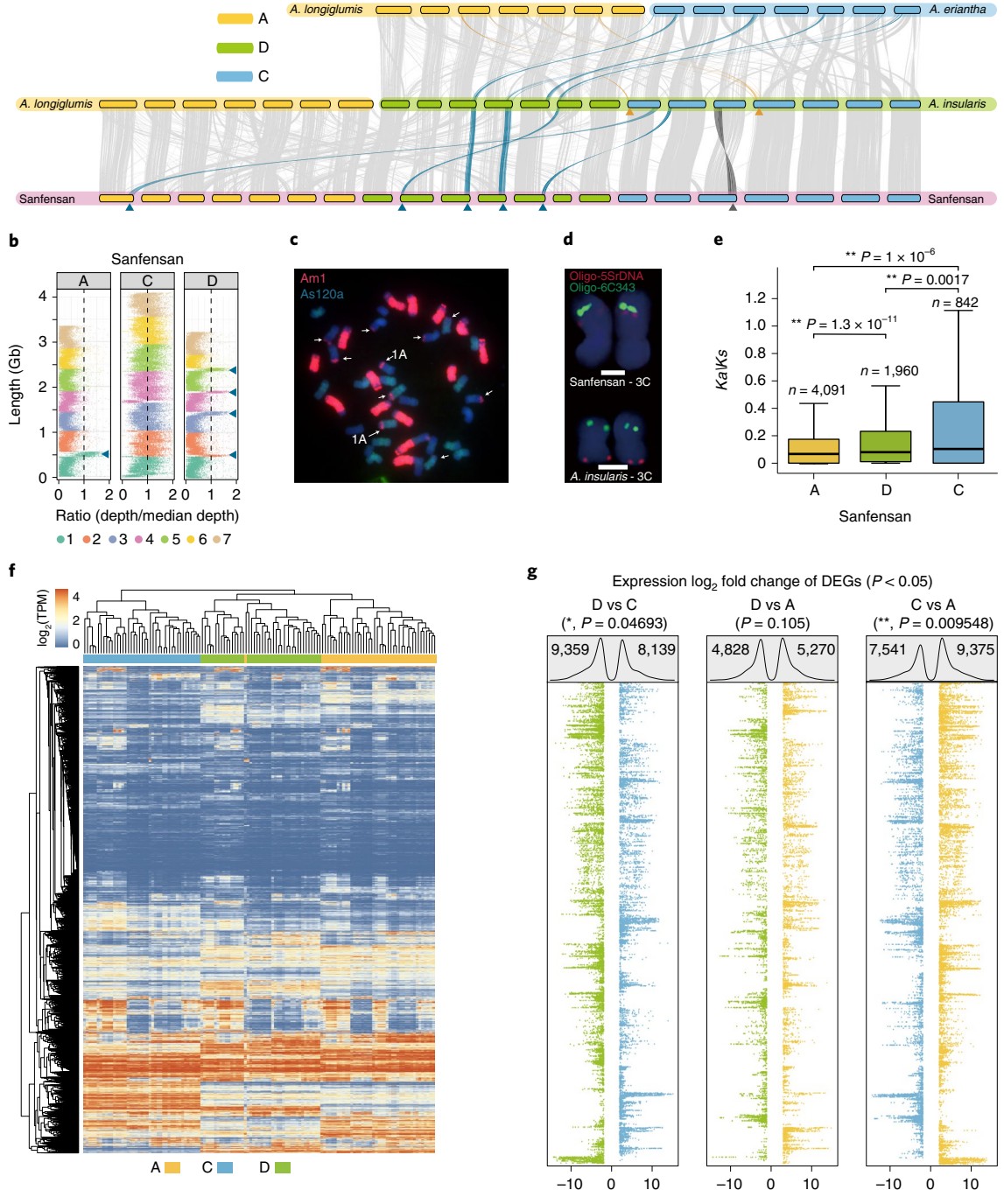

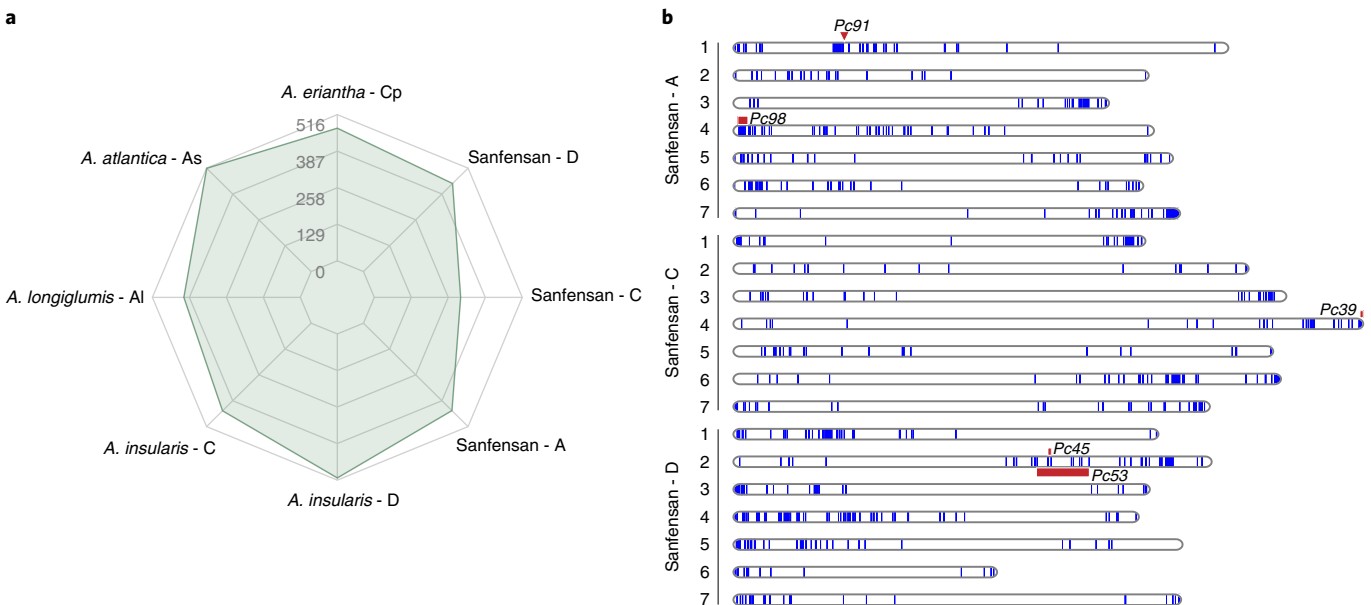

**Fig. 5 | R-genes identified in *Avena* genomes, and distribution on each chromosome of Sanfensan. a**, Comparison of the number of predicted R-genes identified in the genomes of hexaploid oat and its putative progenitors. **b**, Genomic distribution of the R-genes and the location of known quantitative trait loci for crown rust resistance.

Data Fig. 5). To investigate the chromosomal rearrangement events that occurred during the evolution of polyploid oats, we conducted a comprehensive synteny analysis among the diploid, tetraploid and hexaploid species (Fig. 4a,b, Extended Data Fig. 6a–d and Supplementary Tables 15 and 16). The results show that two large D-to-C and four C-to-D chromosomal translocations (>40 Mb) occurred during the formation of tetraploid *A. insularis*. A translocation from chromosome 1C to 1A (>100 Mb) and an inversion in 3C (>130 Mb) subsequently occurred in the hexaploid Sanfensan genome; six of these structural rearrangements were further confirmed by FISH (fluorescence in situ hybridization) assays (Fig. 4a–d, Extended Data Fig. 6 and Supplementary Table 17). Chromosomal rearrangements have been implicated as one of the driving forces for shaping adaption and speciation by affecting gene expression through position effects[25–27]. In oat, the 1C/1A translocation (previously designated as 7C/17A) is well known to be associated with the division of cultivated oat into *A. sativa* L and *A. byzantina* K. Koch (sub)species[28] and variations in crown freezing tolerance and winter field survival[29,30]. Functional enrichment analyses showed that 'multicellular organism development' (GO:0007275, 37/112, BH-Adjusted $P = 9.69 \times 10^{-12}$), 'ubiquitin-dependent protein catabolic process' (GO:0006511, 51/218, BH-Adjusted $P = 4.84 \times 10^{-9}$), and 'oxidation-reduction process' (GO:0055114, 373/3,273, BH-Adjusted $P = 3.00 \times 10^{-6}$) are the top three most enriched biological processes terms for genes positioned within the six large intergenomic translocations that occurred during tetraploidization (Supplementary Table 18). Genes positioned within the 1A/1C translocation were significantly enriched for two biological process terms 'photosynthesis, light harvesting' (GO:0009765, 11/139, BH-Adjusted $P = 2.92 \times 10^{-2}$) and 'regulation of systemic acquired resistance' (GO:0010112, 5/23, BH-Adjusted $P = 2.92 \times 10^{-2}$) (Supplementary Table 19). These processes are essential for responses to abiotic and biotic stress and thus might be involved in local environmental adaption and (or) speciation in oat.

Unlike the chromosomal translocations in the tetraploid *A. insularis* (3.91%, 99.64/2,549.33 Mb), we found that 49.69% (1,054.29/2,121.61 Mb) of the translocated sequences in the hexaploid occurred between homoeologous chromosomes

(Supplementary Tables 15 and 16 and Extended Data Fig. 7). These results suggest that the homoeologous rearrangements after hexaploidization played an important role in forming the genome structure of cultivated oats. Moreover, the homoeologous rearrangements in hexaploid oat appeared to be biased among the three subgenomes in that 88.4% (931.94/1054.30 Mb) occurred between the A and D subgenomes, which is much higher than that occurred in A and C (11.2%, 117.71/1054.30 Mb) or D and C (0.04%, 4.66/1054.30 Mb). This finding may support the hypothesis that the presence of two well-conserved homologous genomes in the same nucleus would facilitate inter-subgenome recombination and rearrangement after polyploidization[31].

Assembly results showed that the C subgenome is ~20% larger than the A and D subgenomes, which can be explained almost entirely by the relative abundances of repeat (Supplementary Table 6). Intriguingly, we observed biased gene fractionation among the subgenomes of hexaploid oat. First, the identification of presence/absence variations (PAVs) (lost genes in tetraploid C vs D, 1,618 vs 1,315, degrees of freedom (df) = 1, $P = 4.13 \times 10^{-9}$; hexaploid C vs A, 2,367 vs 790, df = 1, P < 2.2 × 10⁻¹⁶; and C vs D, 2,367 vs 1,079, df = 1, P < 2.2 × 10⁻¹⁶; chi-squared test) and pseudogenes (C vs A, 28,056 vs 23,695, df = 1, P < 2.2 × 10⁻¹⁶; C vs D, 28,056 vs 24,009, df = 1, P < 2.2 × 10⁻¹⁶; chi-squared test) consistently showed a higher rate of gene loss in the C subgenome of hexaploid oat (Extended Data Fig. 8a–c). More contracted gene families in this subgenome (C vs A: 1,100 vs 569; C vs D: 1,100 vs 535) also support this result (Extended Data Fig. 8d–f). The contracted gene families are mainly related to 'fructose 6-phosphate metabolism', 'the glycine metabolism' and 'translational termination' (Supplementary Table 20). Second, using barley as the outgroup, we estimated the nonsynonymous to synonymous substitution rate ratios ($Ka/Ks$) of the 7,353 single-copy orthologs identified between the oat (sub)genomes. The results showed that the polyploids exhibited an accelerated rate of evolution relative to their A- and C-genome diploid progenitors and that the C subgenome was subject to less purifying selection than the other two subgenomes of hexaploid oat (Fig. 4e and Extended Data Fig. 9). Third, examination of the orthologs expression patterns showed that the number of preferentially expressed

genes in the C subgenome was significantly lower than that in the A (up-regulated genes in A vs C, 9,375 vs 7,541, $P = 0.009548$, Wilcoxon rank-sum test) and D (D vs C, 9,359 vs 8,139, $P = 0.04693$, Wilcoxon rank-sum test) subgenomes (Fig. 4f,g and Supplementary Table 21). Interestingly, these expression-biased genes have more AS variants than those with balanced expression (Extended Data Fig. 10a). For instance, 725 genes were preferentially expressed in the A subgenomes. Of them, 307 have AS variants with an average of 3.10, which was significant more than that of their homeologs in the C (235 genes have AS variants, mean AS variants of 2.77, $P = 0.018$, Student's $t$ test, df = 540) and D (246 genes have AS variants, mean AS variants of 3.70, $P = 0.0036$, Student's $t$ test, df = 551) subgenomes. These results indicated that biased expression patterns may be closely related to the proteome diversity. Fourth, we found that the C subgenome contained more transposable elements (TEs) (A vs C vs D, 2.88 Gb vs 3.61 Gb vs 2.76 Gb) (Supplementary Table 6) and showed a higher overall TE density near genes than did the A and D subgenomes (C vs A, 0.454 vs 0.438, $P = 4.595 \times 10^{-9}$; C vs D, 0.454 vs 0.437, $P = 4.595 \times 10^{-9}$, Student's $t$ test, df = 24,448) (Extended Data Fig. 10b). We further found that genes with relatively higher TE densities near genes tend to have lower expression levels (Extended Data Fig. 10c). This observation is consistent with the hypothesis that subgenome gene expression dominance is influenced by TE-density differences between subgenomes as observed in other allopolyploids[7,32]. Taken together, these results demonstrate subgenome dominance in hexaploid oat.

**Genes related to important agronomic traits.** Oat production is threatened by several agricultural diseases, one of which is crown rust caused by the basidiomycete fungus *Puccinia coronata* var. *avenae*. Nucleotide-binding site leucine-rich repeat (NBS-LRR) proteins, which are encoded by a class of resistance genes (R-genes), play important roles in plant immune signaling[33]. We identified 1,269 R-genes across the three subgenomes of Sanfensan, showing a contraction compared with the numbers identified in the tetraploid and diploid (sub)genomes (Fig. 5a). Most of these R-genes occur in clusters at the distal ends of all chromosome arms. Mapping DNA markers cosegregating with or flanking the known crown rust genes (Supplementary Table 22) revealed these genes colocalize with R-genes (Fig. 5b). These discoveries could serve as promising candidates for future mapping and gene cloning studies.

Cultivated oats are generally classified into two production-related morphological types, hulled and hulless, which is one of the domestication traits (Fig. 6a). The caryopsis of hulled oat is tightly surrounded by a thick, tough and hard-to-remove hull, whereas hulless oat has a papery hull that is easily threshed when mature[34]. To identify genomic loci contributing to the hulless trait in oat, we performed a genome-wide association analysis based on 49,702 SNPs

from 659 diverse oat accessions (Supplementary Table 23). We first infer the population structure and linkage disequilibrium (LD) pattern of the oat collection. Both principal-component analysis (Fig. 6b) and neighbor-joining tree analysis (Fig. 6c) revealed weak population structure, which is consistent with previous studies[35–37]. Most of the hulless landraces were tightly clustered reflecting domestication bottleneck for hulless oat[34]. The LD decay rate was measured as the physical distance at which the average pairwise $r^2$ dropped to 0.2. Genome-wide LD decay rate of this population was estimated at 2.29 Mb (Fig. 6d), The long-range LD observed in oat is similar to that in other self-fertilizing species such as wheat[38,39] and barley[40,41]. An association scan detected a strong peak on the end of chromosome 4D, which collocated with the previously reported *N1* locus[34,42] (Fig. 6e,f). We searched the vicinity of the peak for plausible candidate genes with particular attention to genes within the genomic region covered by markers flanking the *N1* locus[42] (Fig. 6f), and came across a gene (*A.satnudSFS4D01G000045*) annotated as a receptor-like kinase, which is about 30 kb distant from the most highly associated marker. A homologous gene *AtVRLK1* in Arabidopsis was found to be involved in secondary cell wall thickening[43], and the mutant of the homologous gene *mis2* in rice displayed an open hulled spikelet[44]. Comparison the coding sequences (CDSs) of *A.satnudSFS4D01G000045* identified a SNP in first exon predicted to cause amino acid changes (Fig. 6g). The association of this SNP with the hulless trait was validated by the development of KASP (Kompetitive allele-specific polymerase chain reaction) markers (Fig. 6h). By comparing transcriptome data between 10 hulled and 12 hulless oats, we found *A.satnudSFS4D01G000045* is differentially expressed with hulless oats having higher expression levels ($P < 0.01$, Student's $t$ test) (Fig. 6i). Further examination of the expression patterns in the panicles at different developmental stages revealed this gene is highly expressed during panicle development in Sanfensan (hulless) but is expressed at very low levels in panicles of Ogle (hulled) (Fig. 6j, Student's $t$ test). These results indicated that *A.satnudSFS4D01G000045* may be a promising plausible candidate gene that controls the hulled/hulless trait in oat.

## Discussion

The Sanfensan oat, together with its diploid and tetraploid ancestors reference genomes presented here, constitutes an important community resource for cereal genomics and provides comprehensive and specific insights into the evolutionary history of oat. This study is the most comprehensive phylogenomic analysis of *Avena* to date, as it included samples representing all extant *Avena* genomes and developed the largest number of molecular markers evaluated thus far. Whole-genome sequencing, the de novo assembly of transcriptomes and complete plastid genomes show agreement on conclusion. The proposed model for the chronological formation

**Fig. 6 | Genome-wide scan for the hulless grain trait. a**, The spikelets and kernels of hulled and hulless oats. Scale bar, 1 cm. **b**, Population structure of 659 oat lines. Scatter plots of the first principle component (PC1) versus PC2 and PC1 versus PC3. **c**, Neighbor-joining tree constructed based on the Euclidean distance metrics between taxa. **d**, LD decay across the whole genome. The $r^2$ values of LD were plotted against the physical distance (Mb). LD dropped to 0.2 at 2.29 Mb across the whole genome. **e**, Quantile–quantile plot of the MLM model for hulless grain. Linear regression was used to generate the $P$ values. The dotted lines show the 95% confidence interval for the QQ-plot under the null hypothesis of no association between the SNP and the trait. **f**, Manhattan plot for candidate region association mapping of *N1*. The red horizontal line indicates the Bonferroni-adjusted significance threshold ($-\log_{10} P = 6.70$). The zoomed-in region displays the physical positions of the markers GMI_ES22_c7478_431 and GMI_ES14_c19259_657 flanking the *N1* locus[42], and the mostly associated markers. **g**, Gene structure of the candidate gene *A.satnudSFS4D01G000045*. Green boxes are exons, and lines between the boxes are introns. The location of a SNP that resulted in an amino acid change is shown in the diamond shaped box. **h**, The association between *A.satnudSFS4D01G000045* and the hulless trait was further confirmed by a KASP marker derived from the SNP. HEX, hexachloro fluorescein; FAM, fluorescein amidite. **i**, Expression level of *A.satnudSFS4D01G000045* in RNA samples equally mixed from seven tissue/conditions of 10 hulled and 12 hulless oat lines. The central line for each box plot indicates the median. The top and bottom edges of the box indicate the first and third quartiles and the whiskers extend 1.5 times the interquartile range beyond the edges of the box. **j**, Comparison of expression levels of *A.satnudSFS4D01G000045* in panicles at different developmental stages between Sanfensan (hulless) and Ogle (hulled). The data are presented as mean ± s.d. ($n = 3$ biological replicates). S1, S2, S3 and S4 represent panicles at the booting (Zadok's 45), heading (Zadok's 50 and 58) and grain dough (Zadok's 83) stages, respectively. Two-tailed Student's $t$ test was used to generate the $P$ values in panels i and j (*$P < 0.05$, **$P < 0.01$).

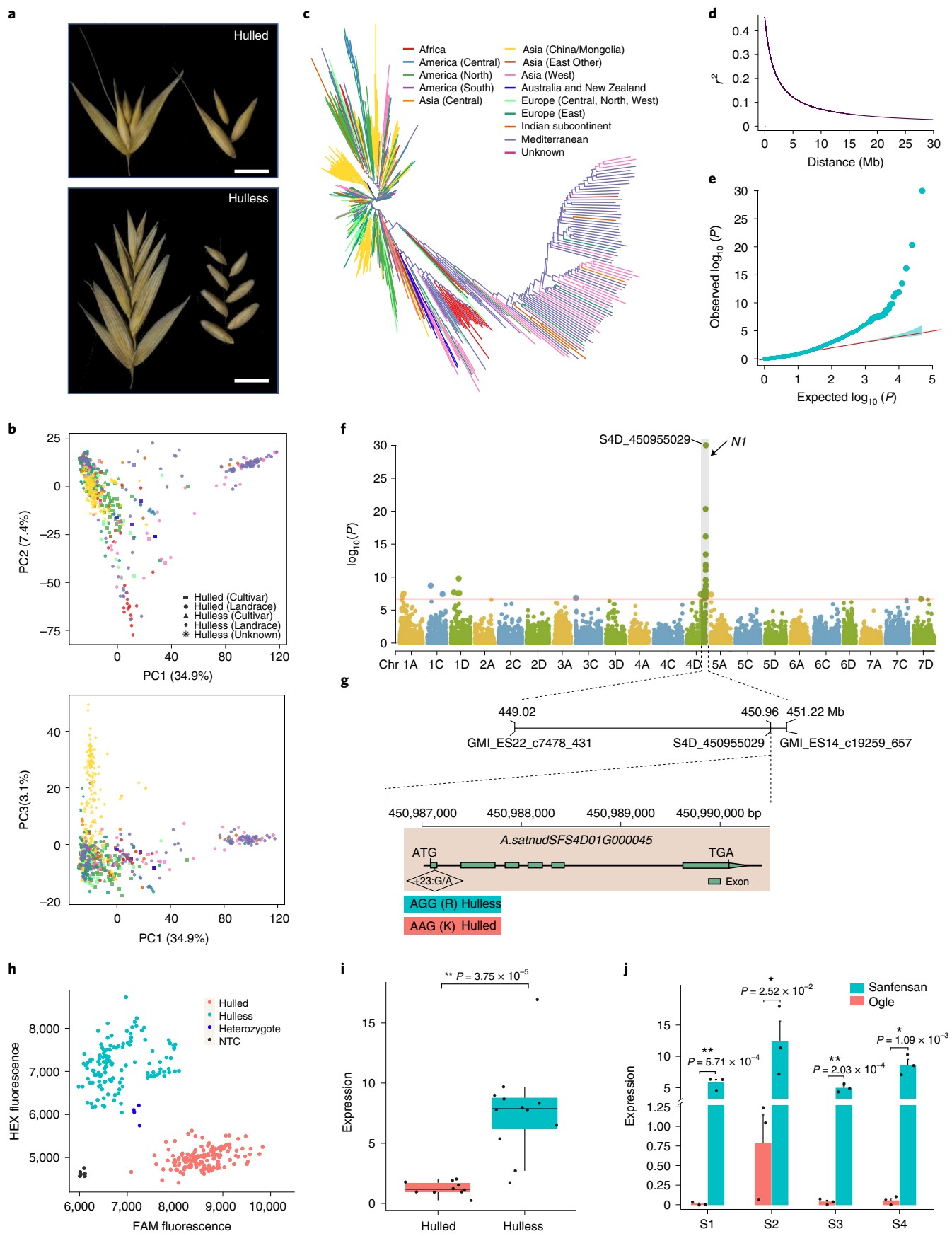

of polyploid oat has been clarified, and the investigation into the evolution of the oat subgenomes during polyploidization events offers a window into the study of polyploid genome evolution. The high-quality reference genomes will facilitate the identification of genes that are related to important agronomic traits, including disease resistance and end-use functionality. These genomic resources are an important step toward the genetic improvement of oat.

## Online content

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

## Methods

**Plant materials.** The hexaploid oat cultivar Sanfensan (2n = 6x = 42, AACCDD), the diploid species *A. longiglumis* (accession CN 58139, 2n = 2x = 14, AlAl) and the tetraploid species *A. insularis* (accession CN 108634, 2n = 4x = 28, CCDD) were selected for genome sequencing and assembly. Sanfensan is a traditional hulless oat landrace that has a long cultivation history in Shanxi, China, which has been widely used as a parental line in hulless oat breeding programs, whereas *A. insularis* and *A. longiglumis* have been considered as the most likely tetraploid and diploid ancestors of hexaploid oat[4].

**Genome sequencing and assembly.** The Illumina HiSeq X-Ten or MGISEQ-2000 platforms were used to generate short paired-end reads from genomic DNA isolated from Sanfensan, *A. insularis*, and *A. longiglumis*. A total of 649.68 Gb, 404.97 Gb, and 204.68 Gb of raw sequencing reads were generated for Sanfensan, *A. insularis* and *A. longiglumis*, respectively (Supplementary Table 1). ONT sequencing was used to generate long nanopore reads for these three taxa. Considering the large and complex genome of hexaploid oat, we used ONT ultralong sequencing to sequence the genome of Sanfensan, whereas ONT regular long reads were generated for the *A. longiglumis* and *A. insularis* genomes. A total of 1,260.30 Gb, 481.39 Gb, and 268.74 Gb of raw ONT (ultra-)long reads were produced for Sanfensan, *A. insularis* and *A. longiglumis* from 71, 7 and 8 libraries, covering their genomes at approximately 100×, 60× and 60× depth, respectively (Supplementary Table 2).

The raw ONT long reads were subjected to self-correction using the NextCorrect module implemented in NextDenovo (v2.0-beta.1) (https://github.com/Nextomics/NextDenovo). The corrected reads were then assembled into contigs using NextDenovo (Supplementary Table 3). To improve the accuracy of the assembled contigs, a two-step polishing strategy was applied: (1) the corrected Nanopore reads were first used for initial polishing with Racon (v1.4.21)[45] (three rounds), and (2) the highly accurate short reads were then used to further correct the assemblies with NextPolish (v1.0.5) (four rounds).

We used Hi-C technology to obtain chromosome-level genome assemblies of Sanfensan and *A. insularis*. A total of 1,312.83 Gb and 816.93 Gb of raw Hi-C data were generated for Sanfensan and *A. insularis*, respectively. The polished contigs of Sanfensan and *A. insularis* were further clustered, ordered and anchored to pseudochromosomes using Hi-C data with the LACHESIS program[46]. We used RaGOO (v1.1)[47] with the default parameters to anchor the contigs of *A. longiglumis* to seven pseudochromosomes with the previously published As genome of the diploid species *A. atlantica*[17] as the reference.

**Genome quality assessment.** Multiple approaches were used to evaluate the quality of the assembled genomes. First, the NG values for all integer thresholds (1–100%) were calculated and used to generate an NG graph for each of the three assembled genomes to estimate the continuity of the assemblies. Second, the short paired-end reads from each species were mapped to their corresponding genome assemblies using BWA (v0.7.10-r789)[48] with default settings. Variants were called using the HaplotypeCaller module of GATK (v4.1.9.0)[49]. The obtained SNPs and InDels were used to estimate the base-level accuracy and relative low heterozygosity of the three assemblies. Third, the BUSCO (v5.2.2)[50] pipeline was used to evaluate the completeness of the gene space. Finally, the LAI[51] was used to evaluate the completeness in the more repetitive genomic regions.

**Subgenome assignment.** A reference-guided strategy based on subgenome sequence similarity was used to distinguish the subgenomes of *A. insularis* and Sanfensan. The *A. longiglumis* reference genome was split into 100-bp nonoverlapping fragments and mapped onto the hexaploid genome assembly using BWA with the default settings. The uniquely mapped fragments were retained (Extended Data Fig. 4b–d). A syntenic block was defined based on the presence of at least five syntenic fragments (Extended Data Fig. 4a,b). The chromosomes with the highest similarity to *A. longiglumis* were assigned to the A subgenome, and the chromosomes with the lowest similarity were assigned to the C subgenome, because previous studies have shown that the C genome is highly diverged from the A and D genomes[4,5]; the remaining chromosomes with median similarity were accordingly assigned to the D subgenome (Extended Data Fig. 4a). Using the same method, the *A. insularis* genomic sequences were aligned to the Sanfensan reference genome, and they showed a high level of similarity with the hexaploid chromosomes that were assigned to the C and D subgenomes (Extended Data Fig. 4b,d). The subgenome assignments were further validated by the quantified depth of coverage of the paired-ended reads from *A. longiglumis* and *A. insularis* t (Extended Data Fig. 4e,f) and the abundance of two types of DNA repeats in the *A. insularis* and Sanfensan genomes, which have been reported to be overrepresented in the *Avena* C[18] and A[19] genomes, respectively (Fig. 1).

**Repeated sequences and protein-coding gene annotation.** Each of the three whole-genome assemblies was searched for repetitive sequences including tandem repeats and TEs. Tandem repeats were identified by using GMATA (v2.2)[52] and Tandem Repeats Finder (v4.07b)[53]. A species-specific de novo repeat library was constructed using MITE-Hunter[54], LTR_FINDER (v1.0.5)[55], and RepeatModeler (v2.0.1) (https://github.com/Dfam-consortium/RepeatModeler). RepeatMasker

(v1.331)[56] was then adapted to search for TEs in the reference genome against Repbase (v19.06)[57] and the species-specific de novo repeat library.

Protein-coding genes were predicted using an evidence-based annotation workflow by integrating evidence from transcriptomic data, homologue searches and *ab initio* prediction. Transcriptomic data were generated by performing PacBio full-length transcriptome sequencing using total RNA isolated from mixed plant organs. Raw reads were processed with IsoSeq3 pipeline (https://github.com/PacificBiosciences/IsoSeq) to identify full-length, nonchimeric circular consensus sequences (CCSs). The resulting high-quality CCSs were mapped onto the reference genome for de-redundancy. The nonredundant isoforms were then used to determine the locations of potential intron-exon boundaries using GeneMarkST[58]. Protein sequences from *A. atlantica*, *A. eriantha*, *Brachypodium distachyon*, *Hordeum vulgare*, *Oryza sativa* and *Triticum aestivum* were used as protein evidence. Ab initio gene prediction was performed using GeneMark-ET (v4.0)[59] and AUGUSTUS (v2.4)[60] with two rounds of iterative training. All gene predictions were integrated using the recommended settings of EVidenceModeler (v1.1.1)[61] after removing TE-related genes, pseudogenes and noncoding genes using TransposonPSI (v1.0.0)[62] with the default settings.

Noncoding RNAs, including microRNAs, small nuclear RNAs, ribosomal RNAs and regulatory elements, were identified using the Infernal (v1.1.2)[63] program to search against the Rfam database[64]. The ribosomal RNAs, transfer RNAs and microRNAs were identified using RNAmmer (v1.2)[65], tRNAscan-SE (v2.0)[66] and miRanda (v3.0), respectively. The putative domains and GO terms of the predicted proteins were identified using the InterProScan (v5.22)[67] program with the default settings.

**Gene family analysis.** Sixteen grass species belong to BOP clade (Supplementary Table 7) for which high-quality reference genomes were available, mainly representing the Pooideae subfamilies Aveneae, Lolieae and Triticeae tribes that contain the main cereal crops were used for gene family clustering analyses. Gene families of the 23 subgenomes from 16 species were identified using the OrthoFinder (v2.2.7)[68] program with default settings. The genes that were exclusively found in each tribe (Aveneae, Lolieae and Triticeae) were identified. Significantly overrepresented GO terms in each group were identified using the R package 'topGO' (https://www.bioconductor.org/packages/release/bioc/html/topGO.html).

**Identification of orthologous and homoeologous gene sets.** Using the gene families identified by the OrthoFinder program, 2,237 one-to-one orthologous gene sets were identified for the 23 subgenomes of 16 grass species. We identified 7,353 one-to-one orthologous gene sets for the eight *Avena* (sub)genomes and *H. vulgare* cv. 'Morex'. We also identified 12,225 one-to-one homoeologous gene sets (triads) for hexaploid oat (Supplementary Table 9).

**Phylogenomic analyses of cereal crops.** To investigate the evolutionary position of oat among cereal crops, a phylogenetic tree was constructed using the 2,237 conserved single-copy genes among the 23 subgenomes of 16 grass species. For this purpose, conserved CDS alignments of these single-copy orthologues were extracted by using Gblocks (v0.9b)[69] and were concatenated to generate a supermatrix. The transversion rates at fourfold degenerate (4DTv) sites were extracted from this supermatrix and subject to analysis with RAxML (v.8.2.7)[70] to generate the maximum likelihood tree using the GTR+I+Γ model. Divergence times were estimated using the MCMCTree program in the PAML(v4.7) package[71].

To investigate the chromosomal evolution of the *Avena* genomes, we selected oat diploid, tetraploid and hexaploid species together with wheat to analyze the synteny between these genomes and the rice or barely genome using MCScanX (git-97e74f40)[72] with the default settings, and the identified syntenic blocks were then used to deduce the homologous relationships between rice or barely genes and the protein sequences of *Avena* and wheat. The collinearity between species was identified and plotted using MCScanX (python version) (Fig. 2c).

**The evolution and allopolyploidization history of oat.** For whole-genome resequencing, 14 other *Avena* accessions were chosen, representing all genome types found among extant *Avena* diploid, tetraploid and hexaploid species (Supplementary Table 1). All genome sequencing was performed on an Illumina HiSeq X-Ten instrument generating 400-base paired-end libraries.

The raw reads from the 14 newly sequenced accessions as well as *A. longiglumis* and *A. insularis* were trimmed using Trimmomatic (v.0.40)[73]. Potential polymerase chain reaction duplicates were removed using SAMtools (v1.9)[74]. The genomic variants were identified using the HaplotypeCaller module in GATK (v4.1.9.0), and SNPs that met any of the following criteria were further discarded: (1) SNPs with quality score <50, QD score <2, FS score >60 or MQ score <40; (2) SNPs with more than two alleles; (3) SNPs at or within 5 bp from any InDels; (4) Genotypes with extremely high (greater than three-fold average depth) or extremely low (less than one-third average depth) coverage were assigned as missing sites; (5) SNPs with a minor allele frequency <0.05; and (6) SNPs with missing sample rate >0.3. SNPs in syntenic regions across the A, C, and D subgenomes for each accession were isolated and used for maximum-likelihood (ML) tree construction using the RAxML software with 200 bootstrap replicates. The phylogenetic relationships of

the species related to the A, C, and D lineages were constructed based on A-, C-, and D-type SNPs, respectively. The level of identity between the individual Sanfensan subgenomes and each sequenced accession was calculated by mapping ~1× clean reads of each accession onto the Sanfensan subgenomes.

All sequenced diploid accessions were further subjected to transcriptome sequencing. The raw reads were cleaned with Trimmomatic. *De novo* assembly was performed using Trinity (v2.0.3)[75] with the default settings. CDSs were predicted using TransDecoder (v5.5.0) (https://github.com/TransDecoder/TransDecoder). Proteomes from these diploid accessions was subjected to gene family analysis against *Hordeum vulgare* protein sequences by using OrthoFinder, and single-copy gene families that were conserved in all species were retained for further study. The protein sequences from each conserved gene family were aligned using MUSCLE (v3.8.31)[76] with the default parameters, and the alignments were CDS back-translated into CDS from the corresponding protein alignments. The same methods described above were used for phylogenetic tree construction and divergence time estimation.

The chloroplast genomes of all sequenced taxa were assembled with NOVOPlasty (v3.7)[77] using the clean Illumina short reads (Supplementary Table 13). Another 25 *Avena* chloroplast genomes previously released[22,23] (Supplementary Table 14) were also downloaded. Multiple sequence alignments were performed using MUSCLE, and the identified informative sites were used for phylogenetic tree construction using RAxML with 100 bootstrap replicates under the GTR+I+Γ evolutionary model, where the chloroplast genome sequence from *Triticum aestivum* was used as the outgroup. Divergence times were estimated based on independent rates and the Jukes-Cantor 1969 (JC69) model using the MCMCTree program in the PAML (v4.7) package.

**Synteny and comparative genomics.** The subgenome synteny between Sanfensan and *A. insularis* was analyzed by plotting the positions of homologous pairs in the subgenome pairs within the context of 21 and 14 chromosomes using Circos (v0.69-9)[78] software (Extended Data Fig. 5a,b). The interchromosomal exchanges between *A. insularis* and Sanfensan after polyploidization were analyzed by individually mapping reads from *A. longiglumis* and *A. eriantha* to the *A. insularis* reference genome and reads from *A. longiglumis*, *A. eriantha* and *A. insularis* to the Sanfensan reference genome. The single-base depth coverage of the properly paired reads obtained from the *A. longiglumis*, *A. eriantha* and *A. insularis* mapping was calculated using the Mosdepth (v0.3.0)[79] program. The median depth within a sliding window (window size: 1 Mb, step size: 0.5 Mb) was calculated and plotted along the chromosomes of the reference genome (Fig. 4b and Extended Data Fig. 6a–d).

**FISH analysis.** Major interchromosomal exchanges between the C and D subgenomes of *A. insularis* and Sanfensan were detected using FISH with the A and C genome-specific repeats As120a and Am1 as the probes. The large inversion observed on hexaploid chromosome 3C was validated by FISH using two additional oligonucleotide probes, Oligo-5SrDNA and Oligo-6C343, as previously reported[80], that gave stable fluorescent signals in regions flanking one of the breakpoints of this inversion on hexaploid chromosome 3C. The nucleotide sequences of the FISH probes were given in Supplementary Table 17. Metaphase chromosome preparation and FISH analysis were performed as described in the previous study[81,82]; (Fig. 4c, Extended Data Fig. 6e, f).

**Ka/Ks analysis.** On the basis of the 7,353 one-to-one orthologous gene sets identified among the genome assemblies for *Hordeum vulgare*, we calculated the nonsynonymous (*Ka*) and synonymous substitution (*Ks*) rates for the A-genome (*A. atlantica* and *A. longiglumis*) and C-genome (*A. eriantha*) diploid progenitors of the hexaploid oat, and the subgenomes of *A. insularis* and Sanfensan. For this purpose, the homoeologous gene pair list was used as the input and the protein sequences from each gene pair were aligned using MUSCLE. PAL2NAL(v14)[83] was used to convert the peptide alignment to a nucleotide alignment, and the *Ka* and *Ks* values were computed between gene pairs using Codeml from PAML (v4.7) in free-ratio mode. All estimates with *Ks* < 0.01 were excluded from the analysis. The significance of the differences in *Ka/Ks* values between genomes (subgenomes) was estimated using the Wilcoxon rank-sum test for nonnormal distributions in R.

**Gene loss and retention.** To measure the effects of polyploidization on gene loss and retention, we performed PAV analyses of the single-copy genes between the polyploids and the diploids. In this case, 14,624 one-to-one orthologues for the three diploids (*A. atlantica*, *A. longiglumis* and *A. eriantha*) were identified. The presence/absence of these genes in each subgenome of the tetraploid and hexaploid species were counted. The distributions of these conserved genes that were absent in the polyploids were displayed with the *A. longiglumis* genome as the reference. Expanded and contracted gene families for eight *Avena* (sub)genomes with the barley genome as reference were identified using CAFÉ (v5.2.1)[84]. Significantly overrepresented GO terms of the contracted gene families in the C subgenome of the hexaploid were identified using the R package 'topGO'. We further compared the gene family size changes in the hexaploid subgenomes and the A- and C-genome diploids based on the gene families identified by the OrthoFinder program. The scatter dots and regression lines were plotted (Extended Data Fig. 8e,f).

**Gene expression analysis.** Gene expression levels in each sample were quantified using the HiSAT2 (v2.2.1)[85] and HTSeq (v0.9.1)[86] pipelines. Differentially expressed genes were identified with the edgeR (v3.38.1) software package[87] (false discovery rate < 0.05 and |log2-fold change > 0.5). To analyze differences in the expression patterns of homoeologous genes, we undertook an initial analysis of the variation in expression among 12,225 homoeologous triplets. Triplet expression vectors were created by concatenating the observed gene expression values for the A, C and D homologs. Triplets that expressed at least one homolog across the sampled tissues were summarized in a triplet expression matrix. The expression values of the triplet expression matrix were log transformed ($\log_{10}$(transcripts per million + 1)), and the matrix was subjected to two-dimensional hierarchical clustering using 'hclust' implemented in R with the 'average' correlation distance and clustering. Heatmap visualization was performed using the heatmap.2 command from the R package gplots.

**Characterization of AS variants.** Transcripts from the RNA-sequencing data were first assembled using StringTie (v2.2.0)[88] with the default parameters based on the bam files generated from HISAT2, which were then integrated with the IsoSeq transcript assemblies using SQANTI3 (v5.0)[89]. The number of AS events in each gene locus was counted using SQANTI3 with the default settings.

**Identification of R-genes.** R-genes in the five *Avena* genomes were identified using InterProScan (v5.22) and DeepCoil (v2.0.1) based on domains from the Pfam (v14.2) database. The domains included coiled-coil (CC), kinase (KIN), Toll/interleukin-1 receptor/resistance protein (TIR), nucleotide-binding site (NBS) and leucine-rich repeat (LRR). Non-canonical R-genes (which lack most conserved motifs but are nevertheless potential R-genes) were determined by BLAST searchers using manually curated R-genes from the PRGdb (v4.0)[90] database as reference sequences. Results with e-values of less than 1E-5 and top-hits with at least 30% query and subject coverage were considered to be the putative R-gene candidates.

To understand whether the identified R-genes were correlated with the map positions of the known quantitative trait loci for crown rust, one of the most serious diseases of oats, DNA markers co-segregating with or flanking the known crown rust genes (Supplementary Table 22) were mapped to the hexaploid Sanfensan reference genome by BLASTn analyses. The distributions of the R-genes and known quantitative trait loci are shown in Fig. 5b.

**Candidate gene for the hulless trait.** A diverse collection of 659 oat accessions that included 510 hulled and 149 hulless oats were used for genotyping by sequencing (GBS) analysis (Supplementary Table 23). Variants were identified using the HaplotypeCaller module of GATK (v4.1.9.0). The resulting variants were further filtered to retain only bi-allelic sites with minor allele frequency > 0.05, missing data rate <20% and heterozygous calls <10% for genome-wide association scans.

Principal-component analysis and neighbor-joining trees were used to infer population structure of the oat collection using TASSEL 5.0 (ref. [91]). Pairwise $r^2$ was calculated using an LD sliding window size of 50. The pattern of LD decay was visualized by plotting pairwise $r^2$ values against the physical distance (Mb). A smoothened LD curve was fit to the data using the R script developed by M. Ali (https://github.com/mohsinali1990/My_scripts/blob/main/LD; decay Plot from TASSEL LDoutput.R).

Genome-wide association scans for the hulless trait were performed with TASSEL 5.0 using a mixed-linear model incorporating the kinship matrix and population structure as covariates. A genome-wide threshold of $-\log(P) = 6.70$, calculated from the formula $-\log_{10}(0.01/\text{effective number of SNPs})$ was used to identify markers associated with the hulless trait.

Ten hulled and 12 hulless oat accession (Supplementary Table 1) were selected for RNA-sequencing analysis. Genomic sequences of the candidate gene from five hulless and five hulled oats were aligned using the ClustalW[92] program. A total of 25 single-nucleotide changes were identified in the gene coding regions between hulled and hulless oats, with one SNP in exon 1 predicted to cause amino acid changes. This SNP was converted to a KASP marker (Supplementary Note) and validated in 286 oat lines randomly selected from the diverse oat collection.

**Reporting summary.** Further information on research design is available in the Nature Research Reporting Summary linked to this article.

## Data availability

The genome assemblies and sequence data for *A. sativa* ssp. *nuda* cv. Sanfensan, *A. insularis* (CN 108634) and *A. longiglumis* (CN 58139) were deposited at NCBI under BioProject codes PRJNA727473, PRJNA731599 and PRJNA716144, respectively. Sanfensan genome assembly (SAMN19770945), ONT data (SAMN19021785), Hi-C data (SAMN19340419), next-generation sequencing (NGS) data (SAMN19582572), IsoSeq data (SAMN19581880) and RNA-seq data (SAMN19582573, SAMN19582574); *A. insularis* genome assembly (SAMN19771048), ONT data (SAMN19291344), Hi-C data (SAMN19312172), NGS data (SAMN19579880) and IsoSeq data (SAMN19581879); *A. longiglumis* genome assembly (SAMN19771099), ONT data (SAMN18395928),

NGS data (SAMN19523931) and IsoSeq data (SAMN19581877). The genotyping-by-sequecing data for 659 oat lines were deposited at NCBI under BioProject code PRJNA807126. All raw data for the other 14 deep-sequenced accessions, including eight diploids, five tetraploids and one hexaploid, are available under the project numbers listed in Supplementary Table 1. Source data are provided with this paper.

## Code availability

The custom codes included in this study are available at GitHub (https://github.com/YuboWang1994/Oat-genome-origin-and-evolution). Codes are also archived at Zenodo (https://doi.org/10.5281/zenodo.6622160) (ref. [93]).

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

## Acknowledgements

This research was supported by China Agriculture Research System of Ministry of Finance and Ministry of Agriculture and Rural Affairs (CARS07 to C.R.), the National Natural Science Foundation of China (32072025 to Yuanying Peng, 31801430 to H.Y. and 31921005 and 31970631 to F.L.), the Science and Technology Development Program of Jilin Province (20200402034NC to C.R.), the Talent Fund Project of Jilin Province, the Fundamental Research Funds for the Central Universities (SCU2021D006 and 2020SCUNL103 to T.M.) and the High-performance Computing Platform of Sichuan Agricultural University. We thank M. Chern, Department of Plant Pathology and the Genome Center, University of California, Davis, for improving the writing of this article. We also thank Agriculture & Agri-Food Canada and E. N. Jellen, Brigham Young University, for providing the *Avena* materials.

## Author contributions

C.R., Yuanying Peng, T.M., F.L. and Yuming Wei conceived the study. C.R., Yuanying Peng, T.M., F.L. and H.Y. provided funding. C.R., Yuanying Peng, H.Y., L.G., P.Z., C.W., J.Z., Yun Peng, D.D. and L.W. collected and prepared the tissue samples for sequencing. Yuanying Peng, T.M., C.D., H.Y., Yubo Wang and F.L. led the bioinformatics analyses.

P.Z., K.Y., C.D., Y.X. and Yun Peng performed the transcriptome sequencing and analysis. K.Y., X.D. and Z.S. constructed the database. X.L. and Ying Li conducted the FISH validation of chromosomal translocations. J.M., M.H. and Yan Li collected pictures of the spikes. C.D., H.Y., Yubo Wang and Yuanying Peng developed the figures. Yuanying Peng, H.Y., L.G., C.D., C.W. and Yubo Wang drafted the manuscript. T.M., L.K., Q.J., J.W., L.W., W.L., H.K., Z.P., D.L., J.J. and Y.Z. contributed to the writing. Yuanying Peng, H.Y., L.G., C.D., C.W. and Yubo Wang contributed equally.

## Competing interests

The authors declare no competing interests.

## Additional information

**Extended data** is available for this paper at https://doi.org/10.1038/s41588-022-01127-7.

**Correspondence and requests for materials** should be addressed to Yuanying Peng, Tao Ma, Yuming Wei, Fei Lu or Changzhong Ren.

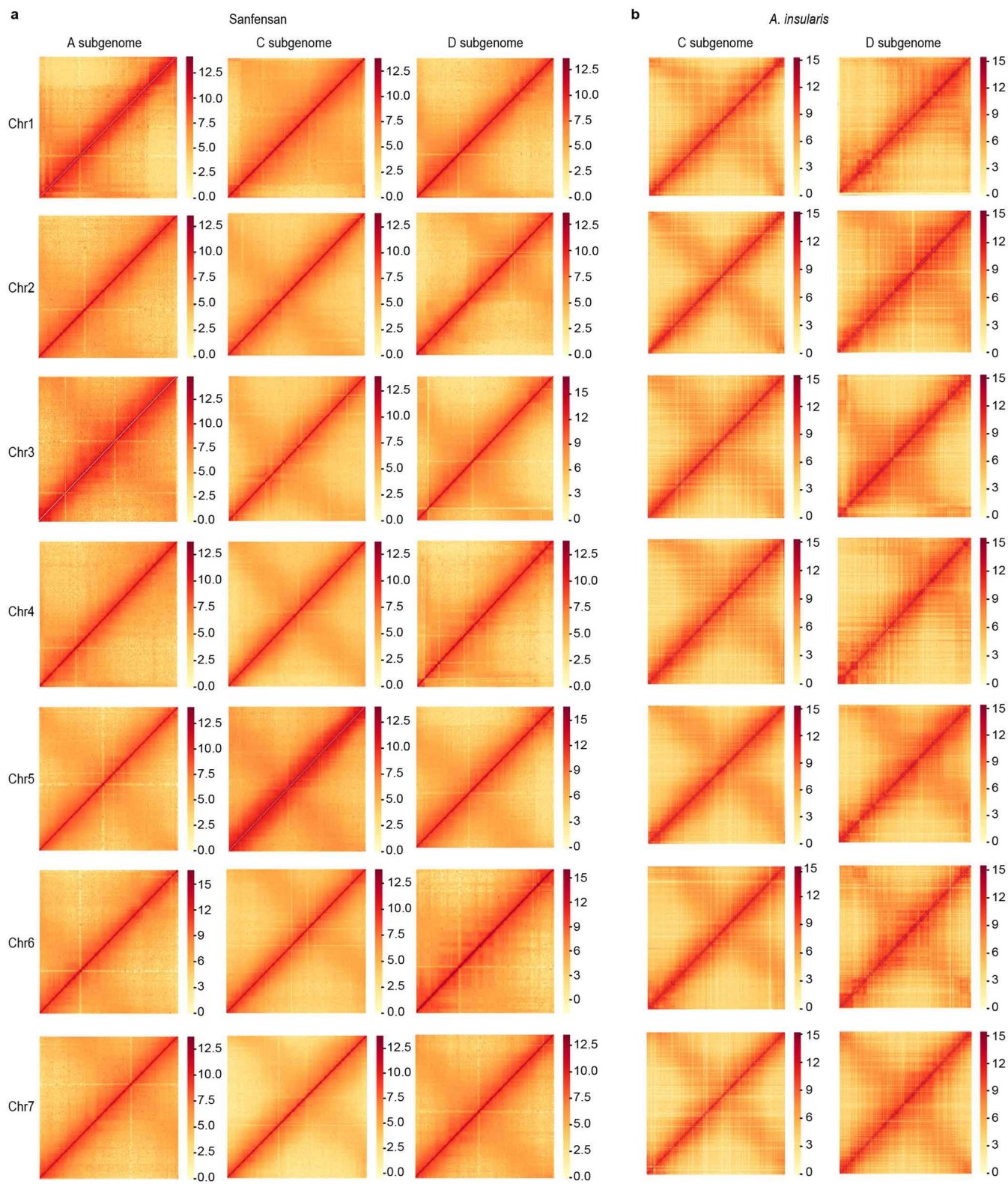

**Extended Data Fig. 1 | Hi-C contact maps for each pseudomolecule in the hexaploid and tetraploid oat genomes. a**, The 21 chromosomes of the hexaploid *A. sativa* ssp. *nuda* cultivar 'Sanfensan'. **b**, The 14 chromosomes of the tetraploid *A. insularis*.

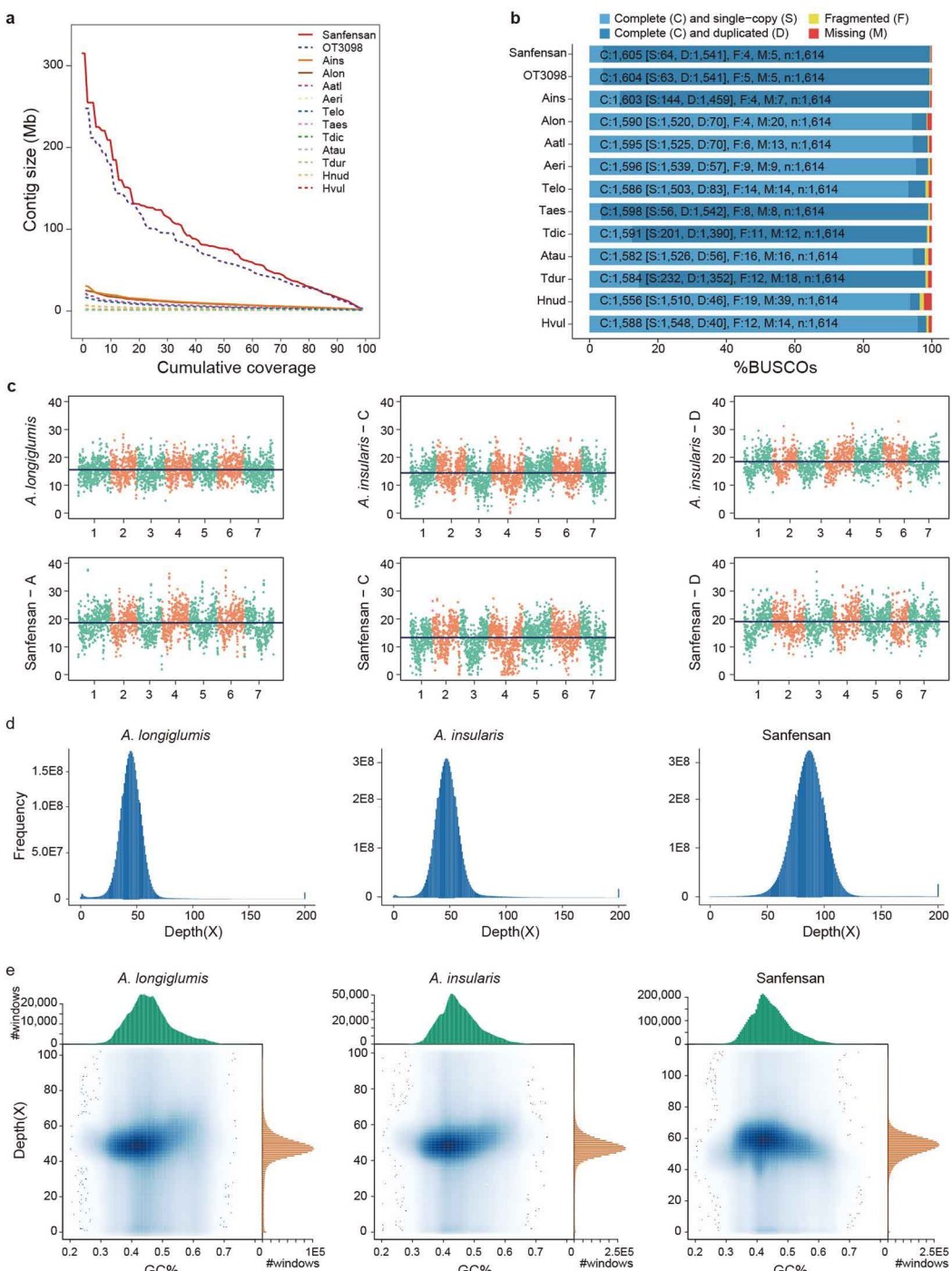

**Extended Data Fig. 2 | Genome assembly quality assessment. a**, NG-graph of the three assembled genomes and other related cereal crop species including *A. sativa* ssp. *nuda* cv. 'Sanfensan' (Sanfensan), *A. sativa* cv. 'OT3098' (OT3098), *A. insularis* (Ains), *A. longiglumis* (Alon), *A. atlantica* (Aatl), *A. eriantha* (Aeri), *Th. elongatum* (Telo), *T. aestivum* (Taes), *T. turgidum* ssp. *dococcoides* (Tdic), *Ae. tauschii* (Atau), *T. turgidum* ssp. *durum* (Tdur), *H. vulgare* ssp. *nudum* (Hnud), and *H. vulgare* cv. 'Morex' (Hvul). The NG contig length is calculated at integer thresholds (1% to 100%) and the contig size (in bp) for that particular threshold is shown on the y-axis. **b**, Completeness of the three assembled oat genomes and the related cereal crop species (as indicated in panel **a**), assessed using BUSCO. Bar charts show the percentages of 1,614 highly conserved plant BUSCO genes that are completely present (light and dark blue), fragmented (yellow), or missing (red) in each assemblies. **c**, The long terminal repeat (LTR) assembly index (LAI) scores of the three assembled *Avena* genomes. The x-axes show the seven chromosomes in each genome. The LAI scores, represented by dots, were calculated using 3 Mb-sliding windows with 1.5 Mb steps. The dark-blue lines indicate the average LAI score across each whole genome. **d**, The distribution of the average depth per million base pairs covered by the ONT reads. **e**, GC content distributions of the three assembled oat genomes. The GC content was determined using 2 kb nonoverlapping sliding windows.

**a**

Linkage group

| | 18 | 33 | 23 | 20 | 24 | 5 | 12 | 28 | 13 | 15 | 11 | 3 | 17 | 9 | 1 | 8 | 19 | 21 | 6 | 4 | 2 |
|---|---|---|---|---|---|---|---|---|---|---|---|---|---|---|---|---|---|---|---|---|---|
| Total | 530 | 425 | 620 | 951 | 787 | 698 | 967 | 601 | 601 | 715 | 780 | 868 | 817 | 772 | 1,258 | 815 | 399 | 1,030 | 727 | 510 | 1,090 |
| 1A | 489 | 2 | 0 | 2 | 0 | 0 | 1 | 28 | 0 | 0 | 1 | 0 | 1 | 12 | 198 | 37 | 0 | 2 | 0 | 0 | 5 |
| 2A | 0 | 364 | 0 | 1 | 0 | 0 | 2 | 0 | 4 | 1 | 0 | 0 | 2 | 0 | 1 | 16 | 0 | 19 | 0 | 2 | 27 |
| 3A | 0 | 0 | 563 | 3 | 1 | 7 | 2 | 14 | 0 | 5 | 0 | 0 | 0 | 1 | 2 | 0 | 4 | 0 | 0 | 0 | 2 |
| 4A | 0 | 1 | 0 | 805 | 1 | 10 | 0 | 0 | 0 | 3 | 3 | 1 | 0 | 5 | 14 | 3 | 1 | 48 | 2 | 1 | 0 |
| 5A | 0 | 0 | 1 | 5 | 499 | 4 | 1 | 0 | 1 | 0 | 1 | 4 | 0 | 1 | 4 | 0 | 0 | 1 | 105 | 0 | 2 |
| 6A | 0 | 0 | 0 | 1 | 0 | 604 | 2 | 0 | 2 | 0 | 32 | 2 | 6 | 0 | 6 | 4 | 0 | 0 | 6 | 22 | 0 |
| 7A | 1 | 1 | 2 | 0 | 0 | 2 | 883 | 0 | 0 | 0 | 0 | 3 | 0 | 0 | 14 | 0 | 5 | 1 | 1 | 0 | 34 |
| 1C | 4 | 0 | 24 | 0 | 0 | 1 | 0 | 545 | 0 | 2 | 0 | 8 | 0 | 11 | 1 | 3 | 0 | 0 | 1 | 0 | 0 |
| 2C | 1 | 4 | 0 | 0 | 0 | 2 | 1 | 2 | 576 | 1 | 0 | 4 | 4 | 2 | 0 | 2 | 0 | 4 | 0 | 0 | 0 |
| 3C | 0 | 0 | 12 | 10 | 0 | 0 | 1 | 2 | 1 | 686 | 27 | 2 | 2 | 3 | 0 | 1 | 12 | 0 | 0 | 0 | 1 |
| 4C | 1 | 0 | 0 | 11 | 3 | 22 | 2 | 2 | 1 | 1 | 684 | 1 | 3 | 0 | 5 | 1 | 1 | 0 | 2 | 0 | 1 |
| 5C | 1 | 0 | 0 | 0 | 4 | 4 | 11 | 3 | 1 | 0 | 0 | 828 | 1 | 2 | 0 | 0 | 1 | 3 | 3 | 0 | 61 |
| 6C | 1 | 5 | 1 | 0 | 0 | 4 | 0 | 1 | 0 | 2 | 1 | 5 | 780 | 0 | 0 | 6 | 0 | 1 | 0 | 16 | 5 |
| 7C | 2 | 0 | 0 | 10 | 1 | 0 | 1 | 0 | 2 | 4 | 1 | 2 | 0 | 658 | 2 | 0 | 0 | 6 | 0 | 0 | 1 |
| 1D | 7 | 2 | 0 | 4 | 46 | 1 | 23 | 0 | 2 | 2 | 3 | 0 | 0 | 1 | 984 | 1 | 7 | 2 | 0 | 0 | 0 |
| 2D | 12 | 8 | 1 | 4 | 0 | 3 | 1 | 0 | 5 | 0 | 1 | 2 | 0 | 0 | 7 | 732 | 0 | 2 | 0 | 0 | 2 |
| 3D | 1 | 0 | 13 | 2 | 0 | 1 | 3 | 0 | 0 | 1 | 2 | 0 | 1 | 0 | 4 | 0 | 366 | 1 | 0 | 1 | 2 |
| 4D | 1 | 2 | 1 | 89 | 2 | 1 | 1 | 1 | 1 | 1 | 1 | 1 | 1 | 33 | 0 | 0 | 0 | 931 | 0 | 2 | 2 |
| 5D | 8 | 0 | 0 | 0 | 224 | 6 | 0 | 2 | 1 | 5 | 7 | 5 | 0 | 0 | 0 | 2 | 0 | 0 | 605 | 0 | 1 |
| 6D | 1 | 0 | 0 | 0 | 0 | 21 | 0 | 0 | 1 | 1 | 0 | 0 | 3 | 0 | 0 | 1 | 0 | 0 | 0 | 465 | 4 |
| 7D | 0 | 14 | 0 | 2 | 0 | 0 | 30 | 0 | 0 | 0 | 2 | 0 | 2 | 0 | 7 | 6 | 0 | 1 | 1 | 1 | 927 |

Sanfensan

**b**

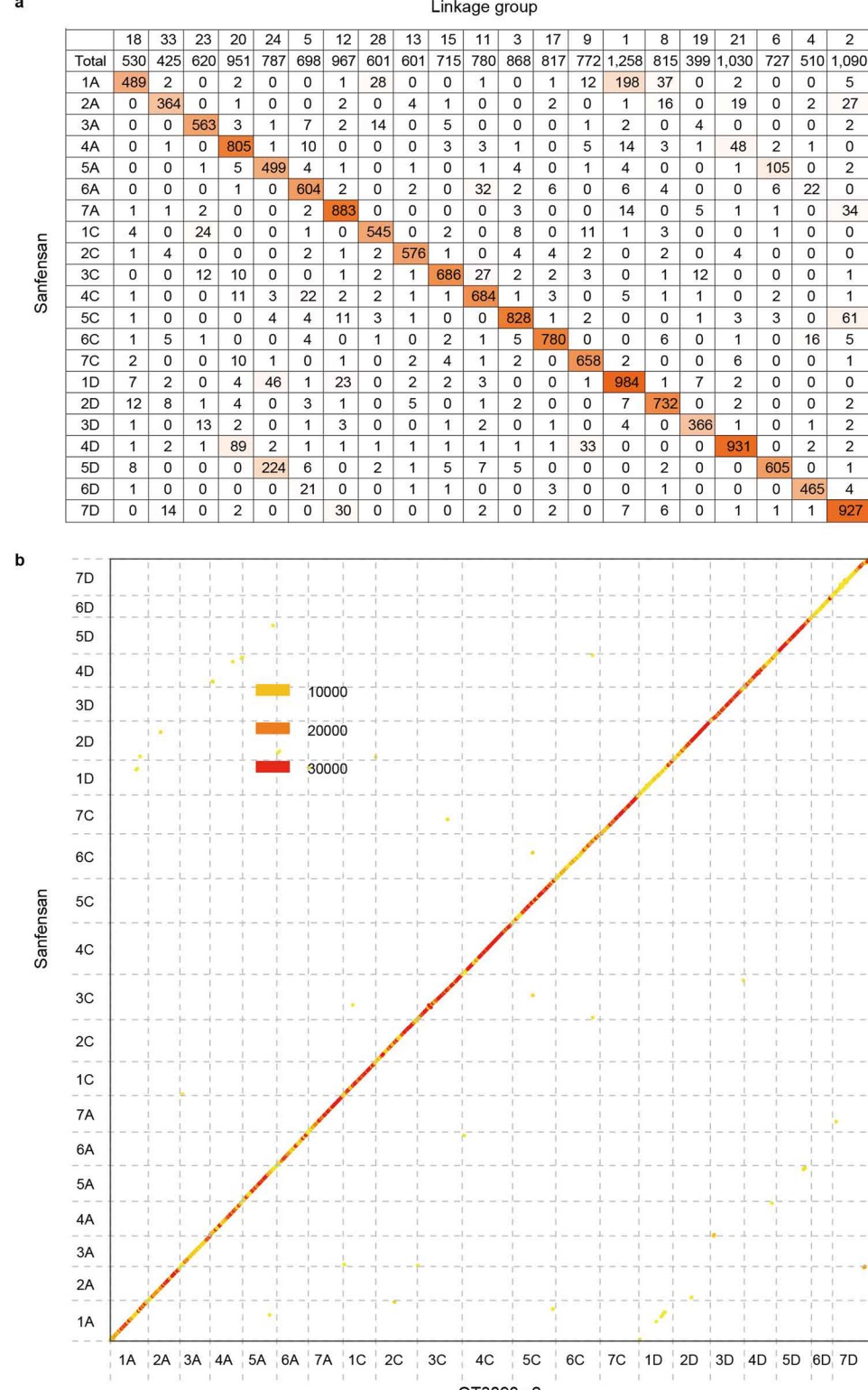

**Extended Data Fig. 3 | Correlation between the 'Sanfensan' genome with the hexaploid consensus map and the OT3098 v2 reference genome. a**, The number of markers from each linkage group of the hexaploid consensus map that are uniquely mapped to the individual chromosomes of the hexaploid 'Sanfensan' reference genome. **b**, Nuclear based comparison between the 'Sanfensan' and hexaploid oat OT3098 v2 reference genomes shows a high level of synteny. The colors indicate the number of fragements in each sytenic block.

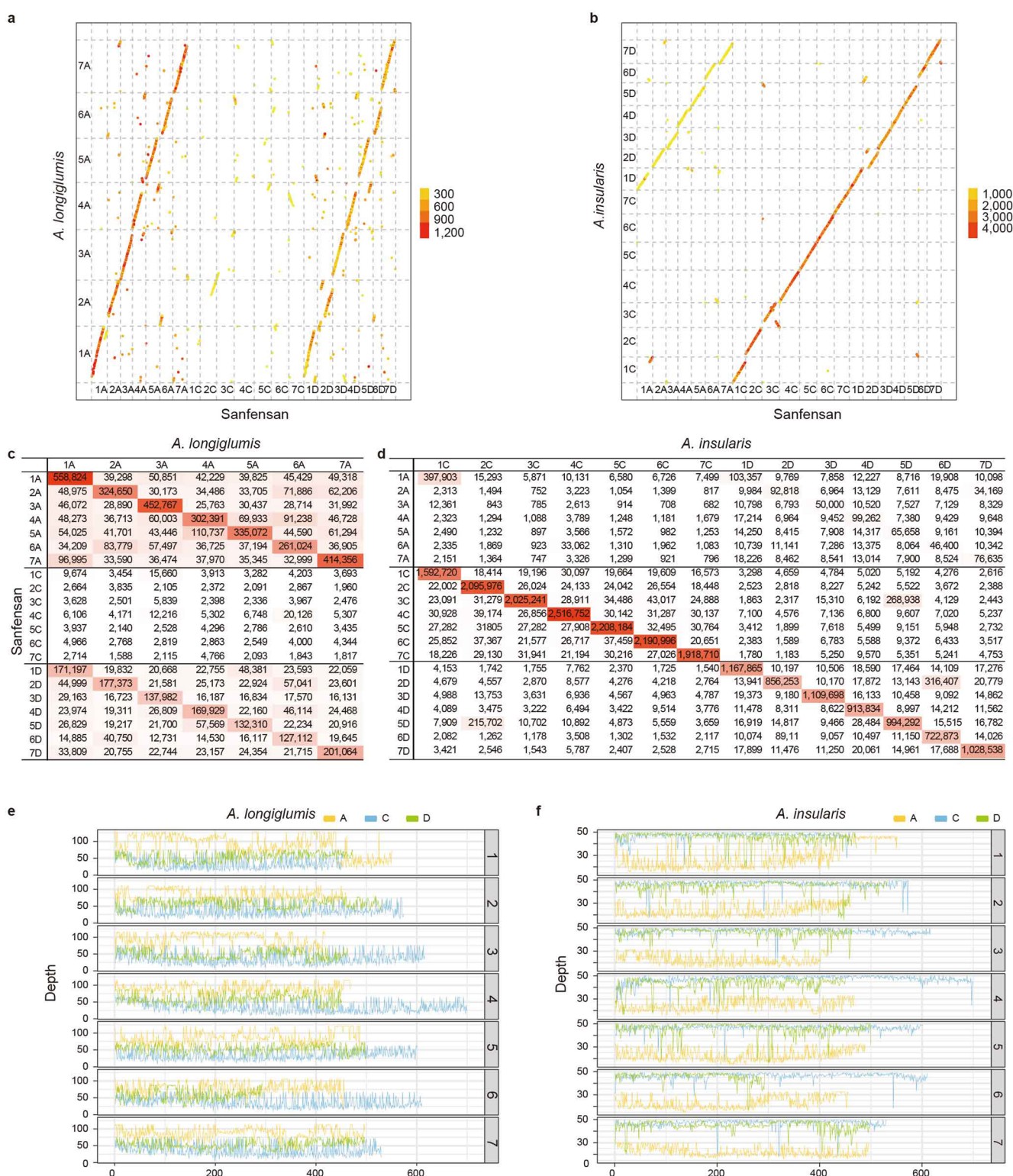

**Extended Data Fig. 4 | Subgenome assignments in hexaploid oat.** The genome sequences of the diploid *A. longiglumis* (Al genome) and the tetraploid *A. insularis* (CD genome) were divided into 100 bp nonoverlapping fragments which were then aligned to the hexaploid 'Sanfensan' reference genome. **a-b**, Dot plots show the distribution of the genomic fragments from *A. longiglumis* (a) and *A. insularis* (b) that were uniquely mapped to the 'Sanfensan' genome. Each dot represents a syntenic block with at least five syntenic fragments. The distance between each pair of adjacent fragments is <200 kb. **c-d**, The number of genomic fragments from *A. longiglumis* (c) and *A. insularis* (d) that mapped uniquely to the individual 'Sanfesan' chromosomes. **e-f**, Coverage depth obtained along the 'Sanfensan' chromosomes after mapping Illumina sequencing reads from *A. longiglumis* (e) and *A. insularis* (f) to the 'Sanfesan' reference genome.

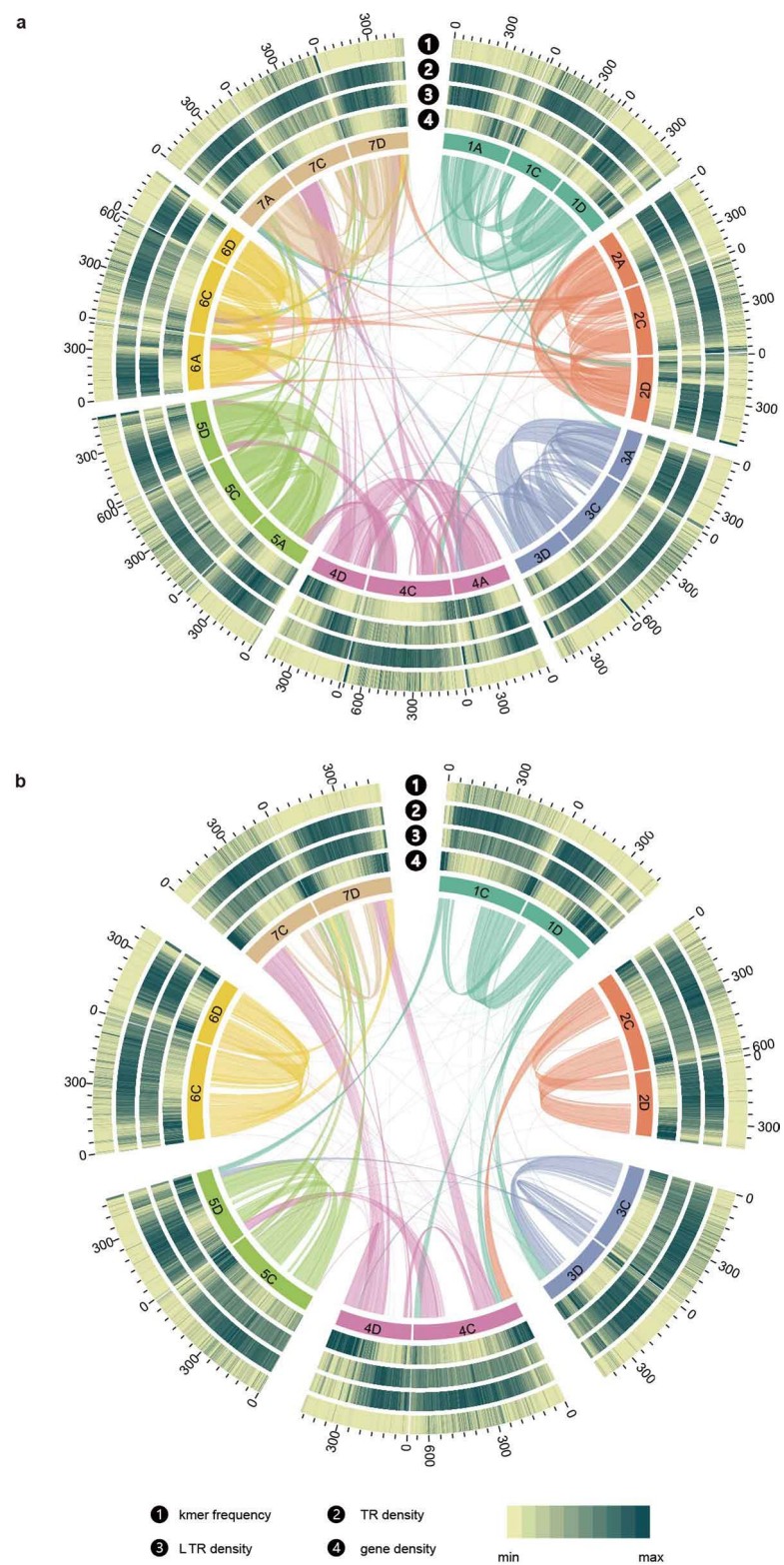

**Extended Data Fig. 5 | Structural and conserved synteny landscape of the 'Sanfensan' (a) and A. insularis (b) genomes.** Homoeologous gene pairs in syntenic blocks (>20 genes) are linked. The four rings depict the 31-mer distribution along each chromosome (1), the density of tandem repeats (TRs, motif length ≤500 bp) (2), the density of long terminal repeats (LTRs) (3), and the density of protein-coding genes (4).

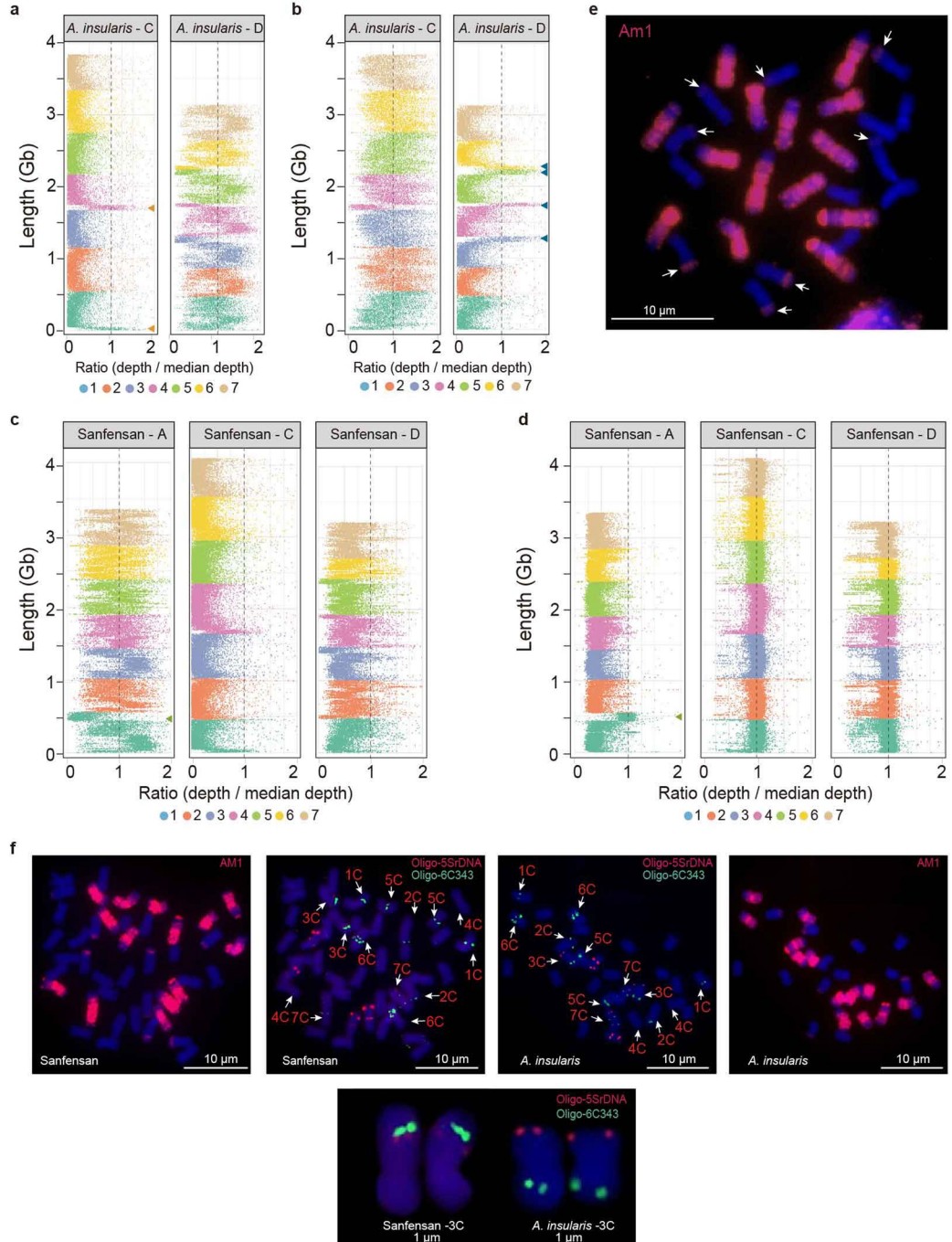

**Extended Data Fig. 6 | Chromosomal structural rearrangements detected in polyploid oats after polyploidization events. a-b**, Mapping short reads of the A -genome diploid *A. longiglumis* (a) and the C-genome diploid *A. eriantha* (b) onto the tetraploid *A. insularis* genome reveals two large (>40 Mb) D/A-to-C and four C-to-D inter-genome translocations. **c-d**, Mapping short reads of the A-genome diploid *A. longiglumis* (c) and the CD-genome tetraploid *A. insularis* (d) onto the hexaploid 'Sanfensan' genome reveals additional large C-to-A intergenomic translocations. **e**, The major C-to-D translocations (indicated by the white arrows) in *A. insularis* were confirmed using FISH technology with the C-genome-specific repeat Am1 (green signals) as the probe. **f**, Confirmation of the inversion on hexaploid chromosome 3 C by FISH. Chromosomes were stained with DAPI (4′,6-diamidino-2-phenylindole). The individual C chromosomes of *A. insularis* and 'Sanfensan' were identified based on the signals generated using Am1, Oligo-5SrDNA, and Oligo-6C343 as probes. Oligo-5SrDNA (red) and Oligo-6C343 (green) gave clear hybridization signals on the short and long arms of the tetraploid 3 C, respectively, whereas both the Oligo-5SrDNA and Oligo-6C343 signals are observed on the long arm of the hexaploid 3 C, suggesting the occurrence of an intrachromosomal rearrangement. For karyotyping, at least three slides for each accession and ten chromosomes per slide were examined.

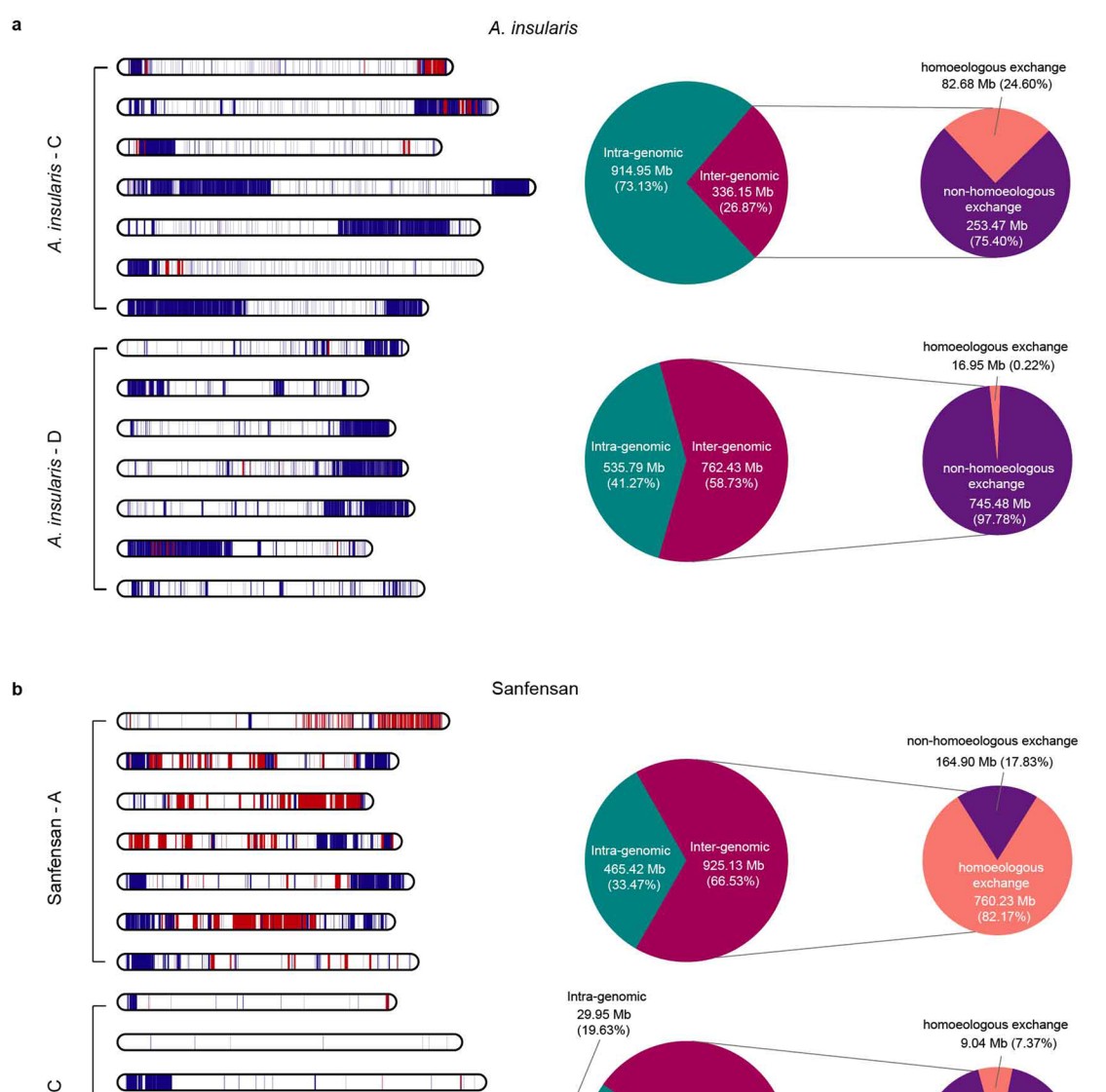

**Extended Data Fig. 7 | Chromosomal rearrangements during tetraploidization and hexaploidization in oat. a-b**, The distribution of homoeologous exchanges (red) and non-homoeologous exchanges (blue) in each chromosome and the proportion of homoeologous exchanges in the C and D subgenomes of *A. insularis* (a) and in A, C, and D subgenomes of 'Sanfensan' (b).

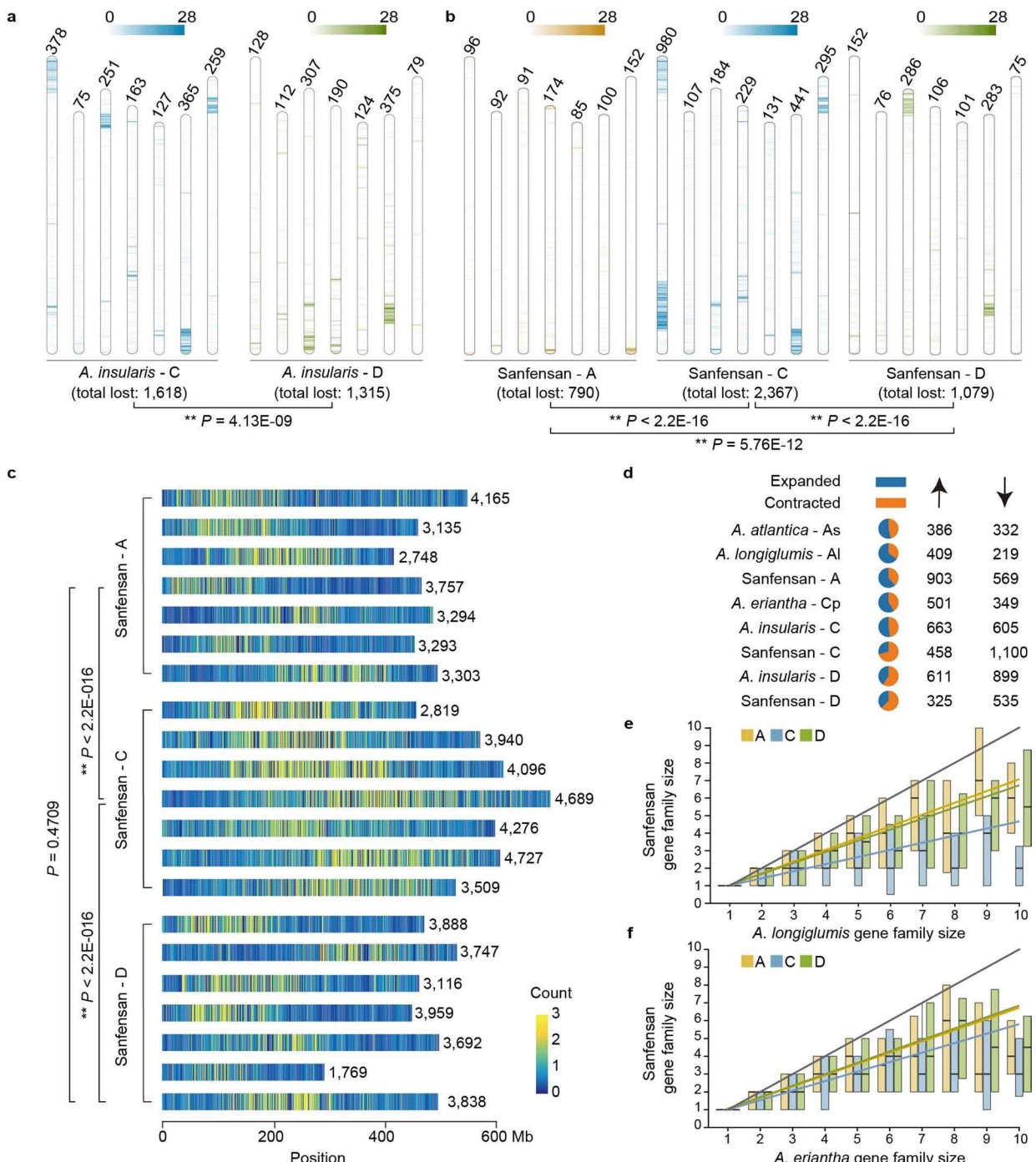

**Extended Data Fig. 8 | Gene conservation and subgenome fractionation patterns. a-b**, Heat maps showing the absence of one-to-one orthologs, which were identified between the A-genome (*A. atlantica* and *A. longiglumis*) and C-genome (*A. eriantha*) diploid species, in the subgenomes of tetraploid *A. insularis* (a) and hexaploid 'Sanfensan' (b). The heat maps were plotted by using the *A. longiglumis* chromosomes as the references, and the number of absent genes was listed for each chromosome. **c**, Distribution and number of the identified pseudogenes along the hexaploid 'Sanfensan' chromosomes. **d**, The numbers of expanded and contracted gene families in the eight *Avena* (sub)genomes. **e-f**, Relationships between gene family sizes in the A-genome diploid *A. longiglumis* (e), and the C-genome diploid *A. eriantha* (f) with each subgenome of the hexaploid 'Sanfensan'. Boxes visualize the median, lower and upper quartiles of gene family sizes. The colored lines show the regression fits for the observed gene family size (yellow, A subgenome; green, D subgenome; blue, C subgenome) and the gray line shows a 1:1 gene copy number relationship for hexaploid oat, *A. longiglumis*, and *A. eriantha*. The two-tailed Chi-square test was used to determine significance in **a**, **b**, and **c** (**, $P < 0.01$).

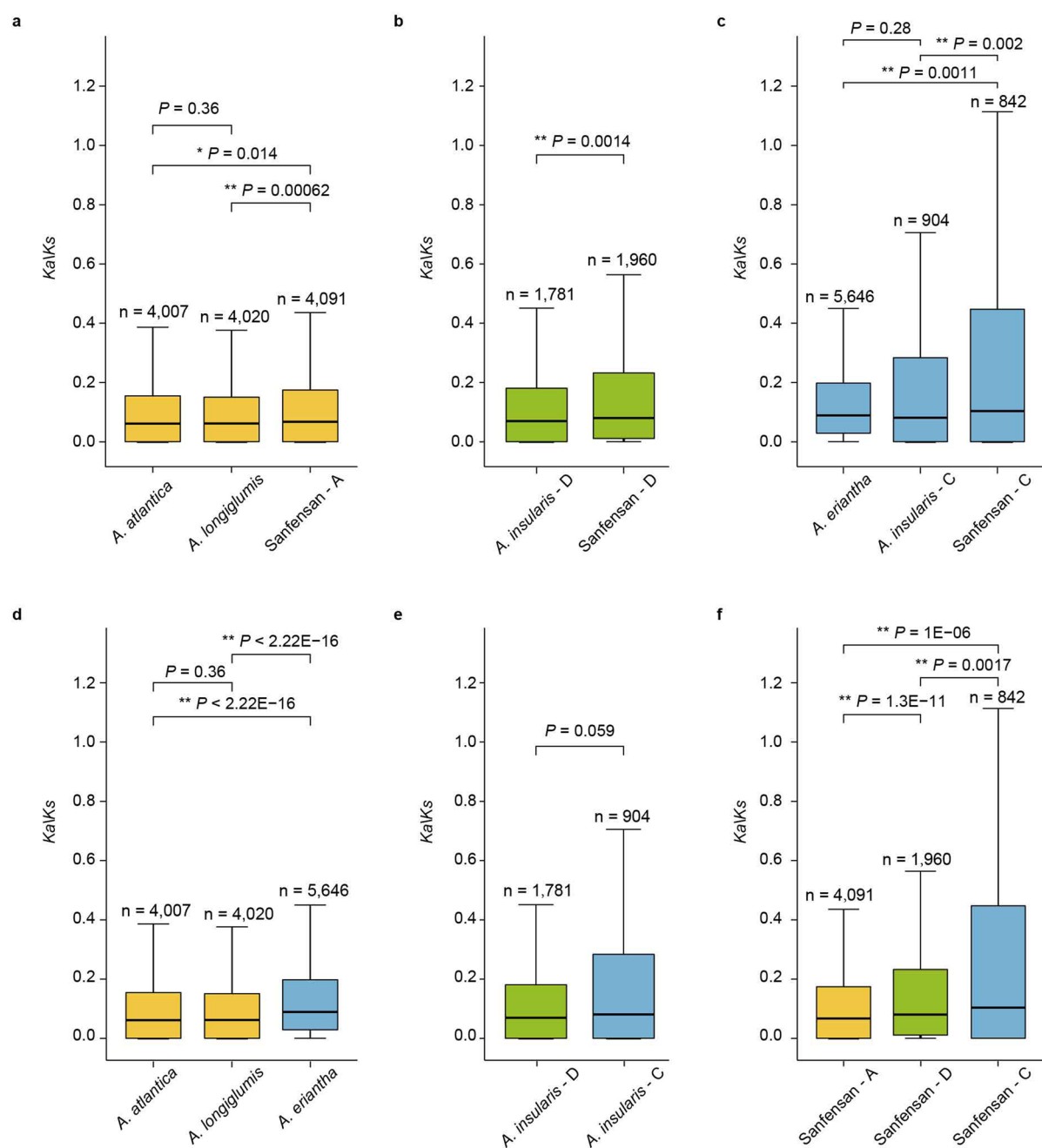

**Extended Data Fig. 9 | Comparison of the *Ka/Ks* distributions between the subgenomes of hexaploid 'Sanfensan' and tetraploid *A. insularis*, and the putative diploid A-genome (*A. longiglumis*, *A. atlantica*) and C-genome (*A. eriantha*) progenitors.** All estimates with $Ks < 0.01$ were excluded from the analysis. The central line for each box plot indicates the median. The top and bottom edges of the box indicate the first and third quartiles and the whiskers extend 1.5 times the interquartile range beyond the edges of the box. Numbers of samples used in each assay are indicated as *n*. The significance of the differences in the values between genomes (subgenomes) was estimated using the two-tailed Wilcoxon rank-sum test (*, $P < 0.05$, **, $P < 0.01$).

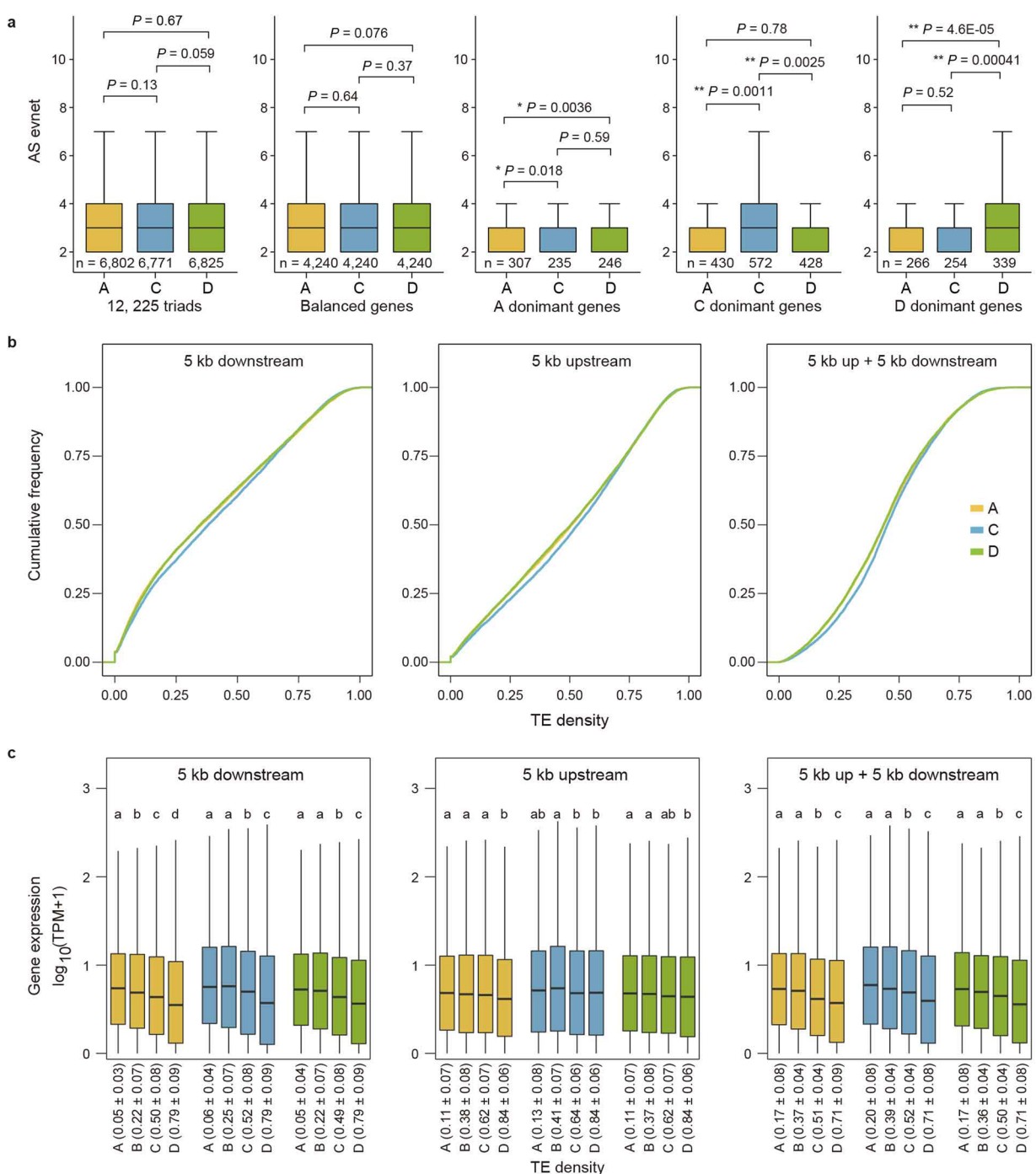

**Extended Data Fig. 10 | Comparison of alternative spicing (AS) events and TE density in the 12,225 strict 1:1:1 triplets in each subgenome of hexaploid oat. a**, The AS number (≥2) of 12,225 homoeologous triads, as well as the AS number (≥2) of genes that showed balanced expression or preferential expression in the A, C, and D subgenomes (left to right) in each subgenome of hexaploid oat. The central line for each box plot indicates the median value. The top and bottom edges of the box indicate the first and third quartiles and the whiskers extend 1.5 times the interquartile range beyond the edges of the box. **b**, Comparison of TE densities near genes in the three subgenomes of hexaploid oat. TE densities near genes in the C subgenome are the highest relative to homoeologs in the A and D subgenomes. TE density was calculated in 5 kb windows upstream and downstream of the gene. **c**, Gene expression is negatively correlated with TE density in hexaploid oat. The 12,225 genes in each subgenome were equally divided into four groups based on their TE density, and their relative expression was then plotted in boxplots. The x-axis represents the TE density in each group. The data are presented as mean±s.d. (n = 9,168, 9,168, 9,168, and 9,171 indepent samples for groups A, B, C, and D, respectively). The central line for each box plot indicates the median value. The top and bottom edges of the box indicate the first and third quartiles and the whiskers extend 1.5 times the interquartile range beyond the edges of the box. Numbers of samples used in each assay in **a** and **c** are indicated as *n*. Two-tailed Student's *t*-test was used to generate the P values in **a** (*, *P* < 0.05, **, *P* < 0.01) and a pairwise *t*-test (two-tailed) was used to determine significance in **c** (means with the same letter are not significantly different at *P* < 0.05).

# Reporting Summary

## Statistics

For all statistical analyses, confirm that the following items are present in the figure legend, table legend, main text, or Methods section.

| n/a | Confirmed | |
|---|---|---|
| ☐ | ☒ | The exact sample size (*n*) for each experimental group/condition, given as a discrete number and unit of measurement |
| ☐ | ☒ | A statement on whether measurements were taken from distinct samples or whether the same sample was measured repeatedly |
| ☐ | ☒ | The statistical test(s) used AND whether they are one- or two-sided *Only common tests should be described solely by name; describe more complex techniques in the Methods section.* |
| ☒ | ☐ | A description of all covariates tested |
| ☒ | ☐ | A description of any assumptions or corrections, such as tests of normality and adjustment for multiple comparisons |
| ☐ | ☒ | A full description of the statistical parameters including central tendency (e.g. means) or other basic estimates (e.g. regression coefficient) AND variation (e.g. standard deviation) or associated estimates of uncertainty (e.g. confidence intervals) |
| ☐ | ☒ | For null hypothesis testing, the test statistic (e.g. *F*, *t*, *r*) with confidence intervals, effect sizes, degrees of freedom and *P* value noted *Give P values as exact values whenever suitable.* |
| ☒ | ☐ | For Bayesian analysis, information on the choice of priors and Markov chain Monte Carlo settings |
| ☒ | ☐ | For hierarchical and complex designs, identification of the appropriate level for tests and full reporting of outcomes |
| ☒ | ☐ | Estimates of effect sizes (e.g. Cohen's *d*, Pearson's *r*), indicating how they were calculated |

*Our web collection on statistics for biologists contains articles on many of the points above.*

## Software and code

Policy information about availability of computer code

| Data collection | No software was used for data collection |
|---|---|
| Data analysis | Guppy (v3.2.2), Jellyfish (v2.0), NextDenovo (v2.0-beta.1) (https://github.com/Nextomics/NextDenovo), NextCorrect, Trimmomatic (v0.40), mininmap2 (v2.18), Racon (v1.4.21), NextPolish (v1.0.5), RaGOO (v1.1), Bowtie2 (v2.3.2), fastp, LACHESIS (https://github.com/shendurelab/LACHESIS), BWA (v0.7.10-r789), SAMtools (v1.9), GATK (v4.1.9.0), BUSCO (v5.2.2), IsoSeq3(https://github.com/PacificBiosciences/IsoSeq), cDNA_Cupcake (v24.3.0) (https://github.com/Magdoll/cDNA_Cupcake ), GMAP (release 2018-07-04), GeneMarkS-T (http://topaz.gatech.edu/GeneMark/license_download.cgi), GeMoMa (v1.6.1), AUGUSTUS (v2.4), BLASTn (v2.7.1), BLASTp (v2.7.1), GeneMark-ET (v4.0), EVidenceModeler (v1.1.1), TransposonPSI (v1.0.0), InterProScan (v5.22), Infernal (v1.1.2), RNAmmer (v1.2), tRNAscan-SE (v2.0), miRanda (v3.0), GMATA (v2.2), Tandem Repeats Finder (v4.07b), MITE-hunter (http://target.iplantcollaborative.org/mite_hunter.html), ClustalW, LTR_FINDER (v1.0.5), LTR_harvest (v1.5.10), LTR_retriever (v2.8), RepeatMasker (v1.331), RepeatModeler (v2.01), TEclass, Repbase (v19.06), Pseudopipe, MACSE (v2), R (v4.05), Mosdepth (v0.3.0), MCScanX (git-97e74f40), MUSCLE (v3.8.31), OrthoFinder (v2.2.7), Gblocks (v0.9b), RAxML (v.8.2.7), PAML (v4.7), FigTree (v1.4.0), CAFÉ (v5.2.1), Trinity (v2.0.3), TransDecoder (v5.5.0), NOVOPlasty (v3.7), Circos (v0.69-9), Photoshop (v7.0), PAL2NAL (v14), HTseq (v0.9.1), HISAT2 (v2.2.1), StringTie (v2.2.0), SQANTI3 (v5.0), DeepCoil (v2.0.1), edgeR (v3.38.1), TASSEL 5.0. The custom codes used to generate the results reported in the study are available at Github (https://github.com/YuboWang1994/Oat-genome-origin-and-evolution/tree/V1.0) and were also archived on Zenodo with DOI: https://doi.org/10.5281/zenodo.6622160. |

For manuscripts utilizing custom algorithms or software that are central to the research but not yet described in published literature, software must be made available to editors and reviewers. We strongly encourage code deposition in a community repository (e.g. GitHub). See the Nature Portfolio guidelines for submitting code & software for further information.

## Data

Policy information about availability of data

All manuscripts must include a data availability statement. This statement should provide the following information, where applicable:

- Accession codes, unique identifiers, or web links for publicly available datasets
- A description of any restrictions on data availability
- For clinical datasets or third party data, please ensure that the statement adheres to our policy

The genome assemblies and sequence data for A. sativa ssp. nuda cv. 'Sanfensan', A. insularis (CN 108634) and A. longiglumis (CN 58139) were deposited at NCBI under BioProject codes PRJNA 727473, PRJNA731599 and PRJNA716144, respectively. 'Sanfensan' genome assembly (SAMN19770945), ONT data (SAMN19021785), Hi-C data (SAMN19340419), NGS data (SAMN19582572), Iso-seq data (SAMN19581880) and RNA-seq data (SAMN19582573, SAMN19582574); A.insularis genome assembly (SAMN19771048), ONT data (SAMN19291344), Hi-C data (SAMN19312172), NGS data (SAMN19579880) and Iso-seq data (SAMN19581879); A.longiglumis genome assembly (SAMN19771099), ONT data (SAMN18395928), NGS data (SAMN19523931) and Iso-seq data (SAMN19581877). The genotyping-by-sequecing data for 659 oat lines were deposited at NCBI under BioProject code PRJNA807126. All raw data for the other 14 deep-sequenced accessions including eight diploids, five tetraploids and one hexaploid are available under project numbers that can be found in Supplementary Table 1. Functional annotation of the genomes used the SwisspProt (ftp.uniprot.org/pub/databases/uniprot/current_release/knowledgebase/complete/uniprot_sprot.fast.gz), NR (ftp.ncbi.nlm.nih.gov/blast/db/FASTA/nr.gz), KEGG (release 97, https://www.genome.jp/kegg/kegg2.html), KOG (ftp://ftp.ncbi.nih.gov/pub/COG/KOG/kyva),GO (http://purl.obolibrary.org/obo/go/go-basic.obo) databases. Non-coding RNA annotation used the Rfam database (http://ftp.ebi.ac.uk/pub/databases/Rfam/14.2/Rfam.tar.gz ). Repetitive element annotation used the Repbase databse (RepBase19.06.embl.tar.gz). The OT3098 v2 (https://wheat.pw.usda.gov/jb?data=/ggds/oat-ot3098v2-pepsico) and hexaploid bread wheat (https://urgi.versailles.inrae.fr/download /iwgsc/IWGSC_RefSeq_Assemblies/v1.1/) reference genomes were retrieved from the GrainGenes database.

# Field-specific reporting

Please select the one below that is the best fit for your research. If you are not sure, read the appropriate sections before making your selection.

☒ Life sciences ☐ Behavioural & social sciences ☐ Ecological, evolutionary & environmental sciences

For a reference copy of the document with all sections, see nature.com/documents/nr-reporting-summary-flat.pdf

# Life sciences study design

All studies must disclose on these points even when the disclosure is negative.

| | |
|---|---|
| Sample size | No statistical methods were required to establish sample size for this study. The Sanfensan cultivar was chosen as a one representative hulless oat since this cultivar has a long cultivated history in China. Tetraploid (A. insularis) and diploid (A. longiglumis) species were chosen as the likely ancestors of hexaploid based on previous marker study. Fourteen additional Avena taxa were used to elucidate the evolution of genus Avena. These Avena taxa represent all genome types found among the extant Avena species. A panel of 659 oat lines from 52 countries or districts were subjected to genotyping-by-sequencing and used subsequently for GWAS analysis. It samples all regions where oat is spread. |
| Data exclusions | No data were excluded from analysis. Raw sequencing data was quality filtered as described in manuscript. |
| Replication | Bootstrapping for phylogenetic analyses based singly copy genes from 23 subgenomes and chloroplast genomes were replicated 100 times, while bootstrapping for phylogenetic analyses based whole genome SNPs were replicated 200 times. Three biological replicates were executed for the RNA-seq and expression analysis. For FISH karyotyping, at least three slides for each accession and ten chromosomes per slide were examined. All attempts at replication were successful. |
| Randomization | Randomizations were not needed for this study, which involved analyzing subgenomes residing within a nucleus of a single genotype. Plants were grown in a sterile growth chamber. |
| Blinding | Group allocation was not relevant to this study, so blinding was not necessary. |

# Reporting for specific materials, systems and methods

We require information from authors about some types of materials, experimental systems and methods used in many studies. Here, indicate whether each material, system or method listed is relevant to your study. If you are not sure if a list item applies to your research, read the appropriate section before selecting a response.

## Materials & experimental systems

| n/a | Involved in the study |
|-----|----------------------|
| ☒ | ☐ Antibodies |
| ☒ | ☐ Eukaryotic cell lines |
| ☒ | ☐ Palaeontology and archaeology |
| ☒ | ☐ Animals and other organisms |
| ☒ | ☐ Human research participants |
| ☒ | ☐ Clinical data |
| ☒ | ☐ Dual use research of concern |

## Methods

| n/a | Involved in the study |
|-----|----------------------|
| ☒ | ☐ ChIP-seq |
| ☒ | ☐ Flow cytometry |
| ☒ | ☐ MRI-based neuroimaging |

