## [Peer Review File · Nature Genetics]

Peer Review Information

Manuscript Title: Reference genome assemblies reveal the origin and evolution of allohexaploid oat

Corresponding author name(s): Dr Yuanying Peng

Editorial Notes:

Transferred manuscripts

This manuscript has been previously reviewed at another journal. This document only contains reviewer comments, rebuttal and decision letters for versions considered at Nature Genetics.

Reviewer Comments & Decisions:

Decision Letter, initial version:

30th Mar 2022

Dear Dr Peng,

Your Article, "Reference genome assemblies reveal the origin and evolution of allohexaploid oat" has now been seen by 2 referees. You will see from their comments below that while they find your work of interest, some important points are raised. We are interested in the possibility of publishing your study in Nature Genetics, but would like to consider your response to these concerns in the form of a revised manuscript before we make a final decision on publication.

We therefore invite you to revise your manuscript taking into account all reviewer and editor comments. Please highlight all changes in the manuscript text file. At this stage we will need you to upload a copy of the manuscript in MS Word .docx or similar editable format.

*2) If you have not done so already please begin to revise your manuscript so that it conforms to our Article format instructions, available [here](http://www.nature.com/ng/authors/article_types/index.html). Refer also to any guidelines provided in this letter.

[REDACTED]

We hope to receive your revised manuscript within four to eight weeks. If you cannot send it within this time, please let us know.

Sincerely,
Wei

Wei Li, PhD
Senior Editor
Nature Genetics
New York, NY 10004, USA
www.nature.com/ng

Reviewers' Comments:

Reviewer #1:

Remarks to the Author:

I reviewed a previous version of this manuscript, but, given the significant changes to the analyses and text, I will treat this as a fresh review.

Peng and colleagues have built an evergreen reference genome for hexaploid oat, which has one of the most complex genomes of any species sequenced to date. Their assembly and annotation, and accompanying genomes and resequencing resources appear to be of the highest quality. I commend their effort. I'll also point out that since Peng and colleague's original submission, the PepsiCo oat genome has become public - the authors acknowledge this. Nonetheless, I strongly support the novelty of the Sanfensan genome presented here. Not only is this genome fairly genetically distinct; but the quality of the genome here will stand up for the future. In my review below, I do not consider the fact that this is not the first public hexaploid oat genome (plus, it kinda was the first, just not public), and I strongly encourage others to follow suit. I have a number of general and line specific comments that I hope will help improve future drafts, which I outline below. I also provide comments regarding the response to reviewers; a few of my previous issues/concerns/comments were not satisfactorily addressed.

- - - [1. Presentation and interpretation of results] - - -

There are many places where upon initially reading the manuscript, the text made it appear that something very different from the actual analysis/method was done. This is a general problem with the presentation of the results/methods/interpretation in the main text, and not a problem with the methods themselves. Here are some examples, but this list is not exhaustive and there are many other places that could be improved:

— [Line 101] how was hic used to correct sequencing errors? I think you mean some unreferenced deep coverage paired end illumina, since it is a line in table 1

— [99/106-112] I'm not following. How did you get from 100X to 20X? In general, I am not following how any of these measures indicate high q genome. For example, why would percent of read pairs have any bearing? I think this is an issue of writing, you mean that the pair map in proximity to a reference, not that they form pairs ... but even then, what is this relative to and how much proximate

3mapping would you need to call a genome high q?

— [136 and elsewhere] How were you able to assign all chromosomes to a sub genome when there are inter-sub genome translocations. This needs to be discussed in other analyses too, like fig 1, where the genome is split into subgenomes. Does this take into account the translocations?

— [228] Are all genes in these regions related to these terms? I think not since it's a go enrichment. Need to describe how much of an outlier these terms are and make sure to discuss any others that are equally or more enriched.

Further, there are several places where there is a logical disconnect between the results and interpretation. Again, I don't see this as a problem with the results themselves, just the presentation of those results. For example: [211-214] I'm not following the logic here. How would knowing the subgenomes are related to extinct species facilitate introgression breeding and transfer of traits. I can think of some ways, but it is not obvious. Please be more precise.

Lastly, I am having trouble understanding the interpretation that translocations affect the "adaptability" of oats, considering that the genes are all still there, just in a different position. Do you have qtl or something showing functional enrichments in these regions. Otherwise, what does adaptability mean, and how were you be able to test that regions contribute to it or not?

- - - [2. Potential issues with the analyses] - - -

There are quite a few places in the manuscript where the analyses appear to not be appropriate for the conclusions the authors draw. For example:

— [114] how many bases were callable? It looks from this stat that the entire genome seq was used as the denominator. Same goes for H0 in the next sentence. Given the large percent of the genome is repetitive, I imagine uniquely mapping reads cover a small percent of the genome.

— [148] why are you using single copy orthologs (i think you mean orthogroups). These will clearly be biased to genes that have sub/non functionalized in your tetraploids, potentially completely biasing all downstream analyses. Especially since your run is with the old orthofinder, which defines orthogroups as those genes that have a single common ancestor among all sampled species. Either drop your polyploids from this run, or use ploidy aware single copy orthologs.

— [266] what is going on here? An R of 0.08 is so significant ... I think you need to contextualize that this is just driven by massive n and very little signal. Further, how would one do anything with such small effect sizes? Some interpretation is important.

Statistics are generally not reported or incomplete. I mentioned this clearly in my previous review, the authors state that they fixed it, but it is far from fixed. For example (again not an exhaustive list):

[218] here and elsewhere, when a comparison is made, statistical support needs to be provided.

[247] no stats again in this whole paragraph

[259] what test, stat and df? Pvalue alone is not sufficient

- - - [3. Phylogenomic analyses] - - -

In the previous draft I reviewed, the authors conducted an across-flowering plant analyses, and it wasn't clear to me why they chose so many species when the proximate goal of all methods in this paper is to understand *Avena* evolution. Although the authors have now restricted the analyses to

4grasses, it remains unclear to me why so many non-avena species are here. No justification is presented and no analyses are explicitly discussed outside of the Avena clade. So, I'll repeat my previous comment with some minor changes: "The purpose of the analysis outside of Avena (and one or two outgroups) was not obvious to me. As a reader, I had trouble grasping how divergence time between (for example) [Sorghum and Maize] integrated with the primary goals of this paper, namely the origin and evolution of the oat genome. How do your assemblies in Avena have any bearing on phylogenetic analysis of other clades? These comments apply to the [poaceae] gene PAV (presence / absence variations) analysis, phylogenetic analyses [143-152] and several other sections."

I still don't understand why you used an ancestral karyotype for any analyses. Why not reconstruct it for Avena and have that as an added value for this paper. I disagree with this statement: "The use of AGK allowed us to trace the history of oat chromosomal structural rearrangements more easily under the framework of phylogeny." This would add error to the analysis, since you could just use barley as the outgroup to phase the three oats. Or maybe this is a problem with communication (see above), and the ancestral oat genome was used, not the ancestral grass genome ... for example [159] is the AGK used here the one from the split between avena and lolium (or maybe barley)? It appears this is not the case and that it is the ancestral karyotype to all grasses, which would add 30M years of error to this analysis. Why did you choose to use this ancestral state that far predated the origin of avena when the only chr evolution discussed is within avena?

Also in my previous review I brought up concerns with the synteny analysis. These still remain, perhaps more so. The authors compared "genomes based on predicted protein sequences using McScan with default parameters." MCScanX with default parameters is designed to look for ancient collinearity and will recover the Rho and other paralog blocks. It appears the authors dealt with this problem by first running the defaults, then culling block to the largest. If this is the case, I am not surprised that there are lots of gaps and overlaps. This is not a good approach and the revised methods do not assuage my previous concerns - look carefully at your block breakpoints ... do they look reasonable? My guess is no, considering the tiny broken blocks in the MCScanX plot and table R1's highly incomplete comparisons.

- - - [4. Minor/specific comments] - - -

— [106] what does transcript % mean?

— [255] what progenitors? I'm not following this logic

— [291] what is the scale of LD in this pop? This is a very wide interval for standard gwas. Either you have truly unparalleled LD or your model is misspecified, perhaps with uncontrolled population structure or non normal phenotypes. The methods appear sound, so not sure what's up here

— [292-293] specific language again. What was examined and why exactly was this gene chosen? The reader needs enough detail to be able to recreate the analysis that resulted in this being the strongest candidate

Reviewer #3:

Remarks to the Author:

The authors have addressed all my requests for additional analysis and clarifications. They have added

5a substantial number of additional analysis and data.

In particular, novel analysis of important agronomic traits is included, adding value to the genomic analysis. Furthermore, a careful comparative analysis to existing data is presented, adding further value.

The manuscript will be of high value for the genomics community.

Author Rebuttal to Initial comments

We appreciate the reviewers very much for the constructive comments and highlighting the contributions of our manuscript. Each suggested revision and comment was accurately incorporated and considered. We have now revised the manuscript according to these valuable suggestions and provide our point-by-point responses. We have made every effort to ensure that our revisions and responses address all of the remaining concerns.

Reviewers'

Comments:

Reviewer	Remarks	to	the	Author:
	I reviewed a previous version of this manuscript, but, given the significant changes to the analyses and text, I will treat this as a fresh review.			#1:

Peng and colleagues have built an evergreen reference genome for hexaploid oat, which has one of the most complex genomes of any species sequenced to date. Their assembly and annotation, and accompanying genomes and resequencing resources appear to be of the highest quality. I commend their effort. I'll also point out that since Peng and colleague's original submission, the PepsiCo oat genome has become public - the authors acknowledge this. Nonetheless, I strongly support the novelty of the Sanfensan genome presented here. Not only is this genome fairly genetically distinct; but the quality of the genome here will stand up for the future. In my review below, I do not consider the fact that this is not the first public hexaploid oat genome (plus, it kinda was the first, just not public), and I strongly encourage others to follow suit. I have a number of general and line specific comments that I hope will help improve future drafts, which I outline below. I also provide comments regarding the response to reviewers; a few of my previous issues/concerns/comments were not satisfactorily addressed.

RESPONSE: We are deeply grateful to these positive and constructive remarks about our study. Specific revisions and responses to each comment are provided in detail below.

6- - - [1. Presentation and interpretation of results] - - -
There are many places where upon initially reading the manuscript, the text made it appear that something very different from the actual analysis/method was done. This is a general problem with the presentation of the results/methods/interpretation in the main text, and not a problem with the methods themselves. Here are some examples, but this list is not exhaustive and there are many other places that could be improved:

RESPONSE: Thanks for raising this point as we realized that our presentation was not sufficiently clear. As suggested, we have checked the manuscript carefully and made improvement on the places that might have not been well presented. Please kindly review all revisions from the attached manuscript text file “marked-up version”. Specific revisions and changes we have made according to your comments are as follows:

— [Line 101] how was hic used to correct sequencing errors? I think you mean some unreferenced deep coverage paired end illumina, since it is a line in table 1.

RESPONSE: Sorry for the unclear statement on this point. The corrections we mentioned here based on the Hi-C data are for the mis-join errors in the raw contigs (i.e., mistakenly assembled contigs), rather than the sequencing errors. As mentioned by the reviewer, the Illumina paired-end reads with higher accuracy were used to correct the sequencing errors. In our study, we generated ~650 Gb of Illumina paired-end reads, representing $\sim 60\times$ coverage of the genome, which were used for the correction of sequencing errors. Whereas the Hi-C data were used to correct the mis-join errors based on three-dimensional proximity information. To avoid misunderstandings, we have added more detailed information about how the genome was assembled and corrected [L92-101] as follows:

“We assembled the ‘Sanfensan’ genome into 326 contigs based on 1,028 Gb of ultralong reads (N50 length: 52 kb; $\sim 100\times$ genome coverage). The draft genome assemblies were then corrected by using ~650 Gb of cleaned Illumina paired-end reads, resulting in a total assembly size of 10.76 Gb, 99.06% of which was arranged into 21 chromosomes (based on 1,296 Gb of Hi-C data) after dissociating 72 mis-joined contigs into 182 contigs using three-dimensional proximity information (Extended Data Fig. 1a). The final assembly contained 436 corrected contigs with N50 of 75.27 Mb and a maximum length of 313.87 Mb. Of these, 323 contigs were anchored onto 21 pseudochromosomes (Table 1, Supplementary Tables 1-3, Extended Data Fig. 1a).”

— [99/106-112] I’m not following. How did you get from 100X to 20X? In general, I am not following how any of these measures indicate high q genome. For example, why would percent of read pairs have any bearing? I think this is an issue of writing, you mean that the pair map in

7proximity to a reference, not that they form pairs ... but even then, what is this relative to and how much proximate mapping would you need to call a genome high q?

RESPONSE: Sorry for the unclear description of the genome assembly quality. The approach we used for genome quality assessment was based on reports for many animal and plant genomes, such as in yak (98% of the assemble was more than 20× sequencing depth, *Nat Genet* 2012 <https://doi.org/10.1038/ng.2343>); rye (the overall alignment rate was 99.77%, *Nat Genet* 2021 <https://doi.org/10.1038/s41588-021-00808-z>) and diploid oat *Avena strigosa* (the overall alignment rate was 98.37%, with 96.33% properly paired alignments, *Nat Commun* 2021 <https://doi.org/10.1038/s41467-021-22920-8>). We mapped the nanopore long reads (~100×) and short pair reads (~60×) to the ‘Sanfensan’ genome and calculated the depth at each base. We found that 99.87% and 96.30% of the assemblies were covered by more than 20× sequencing depth of ultralong reads and short reads, respectively (Table R1 below). On the other hand, the overall alignment rate of the short reads was 99.75%, with 98.22% of these reads correctly pair-mapped to the genome. Altogether, these metrics indicated a high degree of sequence accuracy and continuity of ‘Sanfensan’ genome.

Table R1 | Percentage of genomic regions covered by Nanopore long reads and Illumina short reads

Depth (×)	Base number by long reads covered	Long reads coverage (%)	Base number by short reads covered	Short reads coverage (%)
>1	10,757,092,430	100.00	10,546,501,093	98.02
>5	10,754,194,995	99.97	10,497,301,371	97.56
>10	10,751,423,896	99.94	10,457,590,151	97.20
>20	10,743,833,811	99.87	10,360,888,472	96.30

We have revised the original description to avoid misunderstandings about what is meant when we refer to 20× depth [107-114] as follows:

“In addition, we found that 99.87% and 96.30% of the assemblies were covered at more than 20× sequencing depth with ultralong reads and short reads, respectively, indicating high accuracy at the nucleotide level (Extended Data Fig. 2d-e). Finally, 99.75% of the 4,336,693,678 Illumina paired-end reads could be mapped onto the assembly, with 98.22% properly paired alignments. Based on this mapping, 98,885 homozygous SNPs and 19,444 InDels were identified in ‘Sanfensan’, giving an estimated overall nucleotide accuracy rate of 99.999%.”

— [136 and elsewhere] How were you able to assign all chromosomes to a sub genome when there are inter-sub genome translocations. This needs to be discussed in other analyses too, like fig 1, where the genome is split into subgenomes. Does this take into account the translocations?

RESPONSE: We agree with the reviewer that inter-sub genome translocations do occur in the process of polyploidy, however, the proportion of translocated fragments is not a major part of the whole chromosome (Fig. R1). We can split the polyploid genome into subgenomes based on the genomic similarity among the ACD-genome hexaploid, CD-genome tetraploid and A-genome diploid.

Fig. R1 | Length of individual subgenome DNA sequences inherited from ancestral genomes. Subscript letters represent subgenomes of the hexaploid (H) or tetraploid (T) oat. For example, “C-C_T” means that the DNA in diploid C progenitor remained in the C_T subgenome of tetraploid CD, while “C-D_T” means that the DNA in diploid C progenitor were translocated into the D_T subgenome of tetraploid CD.

We mapped the genomic sequences of the polyploid species to their potential progenitors and calculated the length of sequences uniquely mapped to each chromosome of the progenitors. It can be clearly seen that the sequence similarities between specific chromosome pairs are substantially higher than that between other chromosome pairs (Tables R2 and R3 below, also Revised Supplementary Tables 17 and 18; Figures R2 and R3). These results support identification of orthologous relationships for each chromosome pairs between polyploids and their potential progenitors.

Table R2 | The total DNA length (bp) from each chromosome of *A. insularis* uniquely mapped onto the A and C diploid genomes (Revised Supplementary Table 18).

Chromosome	1C	2C	3C	4C	5C	6C	7C	1D	2D	3D	4D	5D	6D	7D
1C(C5)	350,003,500	73,969,400	694,800	1,434,400	49,500	303,600	90,700	2,853,900	7,191,000	82,834,400	6,800	25,900	70,100	4,300
2C(C4)	64,600	387,693,300	766,600	288,400	98,511,700	2,740,300	2,427,500	2,800	370,200	1,450,900	25,400	44,300	53,700	4,100
3C(C3)	5,327,500	90,900	271,351,700	95,722,600	2,352,800	219,000	803,300	13,072,600	827,500	11,600	2,700	70,383,800	9,500	10,400
4C(C1)	1,371,200	1,377,600	248,700	253,600,900	82,470,500	201,300	197,524,500	23,600	3,600	8,000	3,498,500	366,000	17,100	658,200
5C(C6)	956,100	958,000	953,300	731,600	314,485,300	44,295,800	683,500	1,700	0	313,800	1,100	13,500	39,812,600	15,978,900
6C(C2)	1,198,500	925,200	80,400	5,310,900	158,900	451,779,500	43,954,000	2,300	15,500	6,300	10,956,200	569,200	10,197,400	8,900
C7	26,213,700	33,346,800	65,386,800	118,901,500	71,800	1,793,800	138,220,700	14,508,900	542,400	5,900	58,971,800	8,900	5,600	7,800
1A	37,432,800	33,900	2,000	826,800	5,058,800	1,333,800	2,691,300	327,215,000	8,797,100	20,988,500	1,532,000	5,472,500	36,806,300	24,094,600
2A	0	20,092,000	17,700	11,800	3,000	27,300	1,200	991,900	275,514,300	1,604,900	2,105,200	1,868,300	95,369,000	1,813,300
3A	25,000,400	16,194,100	11,268,800	17,876,300	0	69,200	0	5,594,900	2,775,200	313,237,400	9,937,600	5,560,700	382,400	5,774,500
4A	23,200	4,658,100	2,002,500	177,100	24,136,400	3,700	64,261,300	2,280,500	6,055,800	37,500	254,022,700	46,668,200	4,814,100	7,480,100
5A	56,300	1,200	0	16,738,900	59,100	94,800	9,800	50,394,600	4,104,500	4,750,900	2,473,200	274,412,400	7,279,700	7,221,500
6A	137,200	15,733,300	102,500	33,307,400	8,300	13,638,900	1,200	4,170,400	59,185,200	1,303,000	55,488,500	4,950,500	179,557,000	1,756,500
7A	10,600	17,800	47,600	5,171,100	8,912,200	8,885,000	15,700	5,256,300	2,623,200	1,599,900	7,735,600	5,592,500	11,103,300	411,398,400

*The first row represents the chromosomes of *A. insularis*, the first column represents the chromosomes of the *in silico* tetraploid, merged with the diploid A (*A. longiglumis*) and C (*A. eriantha*) genomes.

Table R3 | The total DNA length (bp) from each chromosome of Sanfensan uniquely mapped onto their putative diploid and tetraploid progenitors (Revised Supplementary Table 17).

Chromosome	Sanfensan_1A	Sanfensan_2A	Sanfensan_3A	Sanfensan_4A	Sanfensan_5A	Sanfensan_6A	Sanfensan_7A	Sanfensan_8C	Sanfensan_9C	Sanfensan_10C	Sanfensan_11C	Sanfensan_12C	Sanfensan_13D	Sanfensan_14D	Sanfensan_15D	Sanfensan_16D	Sanfensan_17D	Sanfensan_18D			
A. longiglumis_1A	18,284,000	11,000,000	1,600,000	4,200,000	6,200,000	42,200,000	42,200,000	0	0	2,600	0	0	1,200	22,204,000	9,641,000	42,100,000	1,000,000	0	2,210,000		
A. longiglumis_2A	54,100	244,910,000	1,655,700	1,760,700	539,900	41,435,000	158,400	0	0	2,900	0	0	0	35,745,000	0	2,100	1,280,000	11,622,000	0		
A. longiglumis_3A	3,900,000	3,363,700	144,948,000	24,970,700	1,341,300	19,989,000	2,854,200	13,908,700	1,883,200	3,415,100	0	0	0	5,200	1,764,700	6,346,700	431,400	0	1,443,000		
A. longiglumis_4A	910,000	3,020,800	140,400	149,000,000	86,700,000	3,092,400	3,283,300	0	0	4,200	4,600	10,000	0	1,624,200	2,618,400	0	12,240,000	4,420,000	6,000	1,610,000	
A. longiglumis_5A	4,400,000	940,700	2,070,000	56,600,000	102,200,700	4,102,400	1,154,200	0	0	0	0	0	0	14,800	3,116,100	4,010,000	127,000	1,882,000	5,020,000	4,910,000	
A. longiglumis_6A	1,485,700	42,627,000	0	19,800,000	3,313,000	276,304,000	150,000	181,000	0	14,451,200	0	0	0	2,084,500	3,030,900	177,000	26,472,200	10,100	27,902,000	0	
A. longiglumis_7A	660,400	23,823,000	2,401,800	5,427,300	12,768,000	3,408,400	121,043,000	0	0	0	0	0	14,200	900	3,182,400	0	1,600	1,200	1,116,000	66,144,900	
A. insularis_1C	198,373,000	73,400	29,600	0	0	8,700	413,588,000	172,900	2,520,000	976,400	1,800	0	1,708,100	0	0	0	0	607,200	0	0	
A. insularis_2C	1,570,400	1,200	0	0	0	0	140,370,000	412,200	1,007,700	793,000	1,080,000	401,500	0	1,648,400	0	0	62,904,000	0	0	0	
A. insularis_3C	1,286,500	0	0	0	80,800	0	239,800	72,000	476,312,000	71,300	0	2,400	807,900	0	0	1,431,400	183,200	3,198,200	0	0	
A. insularis_4C	0	0	0	2,600	0	21,476,900	0	1,836,900	60,000	1,100,000	430,632,000	1,738,400	2,290,700	400,900	0	49,000	2,000	660,900	0	118,300	
A. insularis_5C	414,000	0	0	0	408,700	0	686,900	2,300	1,872,100	430,100	589,020,000	2,544,200	1,080,700	0	0	0	4,400	0	0	0	
A. insularis_6C	1,200	23,000	0	0	430,900	0	30,200	0	2,349,400	0	661,700	20,000	700	0	0	41,500	1,319,000	100,000	0	10,700	
A. insularis_7C	793,000	0	0	0	0	0	21,500	0	96,700	976,800	2,568,100	75,400	679,303,000	0	0	0	0	1,700	959,200	0	
A. insularis_8D	6,904,000	1,900	461,500	11,473,000	29,547,500	3,300	32,294,000	174,300	0	511,700	0	1,800	0	308,365,000	6,000	51,800	16,200	246,700	85,100	493,500	
A. insularis_9D	426,200	119,216,000	0	2,800	0	3,642,900	1,177,100	0	0	0	0	0	0	0	0	244,808,000	120,100	7,400	745,200	761,700	219,200
A. insularis_10D	0	0	196,240,000	11,063,400	702,900	0	1,849,900	29,500	305,300	473,000	0	0	1,321,000	0	875,900	0	278,820,000	0	2,200	396,200	0
A. insularis_11D	0	1,800	0	182,060,700	2,400	140,000	14,700	0	24,100	242,000	0	0	1,000	436,900	0	3,500	16,000	261,600,000	0	300,000	
A. insularis_12D	905,000	0	4,046,200	239,500	43,594,900	18,500	0	0	79,084,900	470,000	0	0	5,800	3,800	431,900	0	184,000	1,800	159,797,400	1,700	1,236,000
A. insularis_13D	7,010,000	0	4,200	0	176,338,200	0	0	0	0	0	0	0	0	311,800	821,000	113,016,100	0	872,200	679,000	32,485,000	411,100
A. insularis_14D	0	14,249,000	1,900	0	2,300	1,600	30,123,200	0	38,200	0	0	0	0	0	0	0	138,900	1,200	234,400	1,393,000	49,043,400

*The first row represents the chromosomes of Sanfensan, the first column represents the chromosomes of the *in silico* hexaploid, merged with the diploid *A. longiglumis* genome and the tetraploid *A. insularis* genome.

The percentage of genomic regions of each tetraploid *A. insularis* chromosome that were uniquely aligned to the diploid A and C genomes.The percentage of genomic regions of each hexaploid ‘Sanfensan’ chromosome that were uniquely aligned to the diploid A genome and the tetraploid CD genome.

In addition, previous studies using Southern blots isolated two DNA repeats, namely As120a (*Proc Natl Acad Sci USA* 1996 <https://doi.org/10.1073/pnas.95.21.12450>) and Am1 (*Theor Appl Genet* 1992 <https://doi.org/10.1007/BF00226904>) from the A genome diploid *A. strigosa* and the CD genome tetraploid species *A. murphyi* respectively. The As120a was found to be overrepresented in the hexaploid A genome chromosomes and some A genome diploid chromosomes, while the Am1 is much abundant in all the C genome chromosomes. By using these two repeats as probes, Linares et al., (*Proc Natl Acad Sci USA* 1996) successfully allocated all of the 21 chromosomes into A, C, and D subgenomes. We assessed the abundances and distributions of As120a and Am1 on the ‘Sanfensan’ and *A. insularis* chromosomes: this confirmed that As120a and Am1 are indeed overrepresented on the chromosomes that have been assigned as the A and C genomes respectively (Fig. R4 below, also Revised Main text Fig. 1, circles b and c), thereby, supporting our subgenome assignments.

Fig. R4 | Circos display of the genomic features of the assembled diploid *A. longiglumis*, tetraploid *A. insularis*, and hexaploid *A. sativa ssp. nuda* genomes. **b**, The distribution of the C genome-specific repeat Am1 along each chromosome. The Am1-rich regions on chromosomes 1A, 2D, 3D, 4D, and 5D of ‘Sanfensan’ are C genome introgressions. **c**, The distribution of the A genome-specific repeat As120a along each chromosome (Revised Main text Fig. 1).

— [228] Are all genes in these regions related to these terms? I think not since it’s a go enrichment. Need to describe how much of an outlier these terms are and make sure to discuss any others that are equally or more enriched.

16RESPONSE: Thank you for making this valuable suggestion. Not all genes in these regions are related to these terms. We just primarily summarized the putative biological processes of the top-ranking GO terms maybe involved in the original manuscript. In the revised manuscript, we have listed all enriched terms in the supplementary tables 15 and 16, and provided more details on the top-ranking GO terms including the GO id, gene ratio, and BH-adjusted P value in the main text [L251-261] as follows:

“Functional enrichment analyses showed that ‘multicellular organism development’ (GO:0007275, 37/112, BH-Adjusted $P = 9.69E-12$), ‘ubiquitin-dependent protein catabolic process’ (GO:0006511, 51/218, BH-Adjusted $P = 4.84E-09$), and ‘oxidation-reduction process’ (GO:0055114, 373/3,273, BH-Adjusted $P = 3.00E-06$) are the top three most enriched biological processes terms for genes positioned within the six large inter-genomic translocations that occurred during tetraploidization (Supplementary Table 15). Genes positioned within the 1A/1C translocation were significantly enriched for two biological process terms ‘photosynthesis, light harvesting’ (GO:0009765, 11/139, BH-Adjusted $P = 2.92E-02$) and ‘regulation of systemic acquired resistance’ (GO:0010112, 5/23, BH-Adjusted $P = 2.92E-02$) (Supplementary Table 16).”

Further, there are several places where there is a logical disconnect between the results and interpretation. Again, I don’t see this as a problem with the results themselves, just the presentation of those results. For example: [211-214] I’m not following the logic here. How would knowing the subgenomes are related to extinct species facilitate introgression breeding and transfer of traits. I can think of some ways, but it is not obvious. Please be more precise.

RESPONSE: We thank you for this comment and have revised the interpretation to explicate our intended meaning as follows [L228-232]:

“These findings clarified the evolutionary history of oats based on various pieces of evidence at the genomic level and provided the most likely clues to the subgenomic origin of hexaploid oat, which will be of great value for introgression breeding and the transfer of traits from the closest extant wild relatives (As/Al genome diploids and CD genome tetraploids) to cultivated oat.”

Lastly, I am having trouble understanding the interpretation that translocations affect the “adaptability” of oats, considering that the genes are all still there, just in a different position. Do you have qtl or something showing functional enrichments in these regions. Otherwise, what does adaptability mean, and how were you be able to test that regions contribute to it or not?

RESPONSE: Thank you for focusing our attention here. Many previous studies have indicated that chromosomal rearrangements including inversions and translocations have played pivotal role in

shaping adaptation and speciation by affecting gene expression through e.g., position effects (position of a gene on a chromosome) (*Mol Ecol* 2017 <https://doi.org/10.1111/mec.14442>; *Mol Ecol* 2019 <https://doi.org/10.1111/mec.14923>; *Environ Microbiol* 2019 <https://doi.org/10.1111/1462-2920.14586>). In oat, the 1A/1C (previously designated as 7C-17A) translocation has been demonstrated to be highly associated with the division of cultivated oat into *A. sativa* L. and *A. byzantina* K. Koch (sub)species; in 77 of 97 cases examined *A. sativa* type accessions from various countries have the 1A/1C translocation, while 81 of 91 *A. byzantina* type accessions lack this translocation (*Crop Sci* 2000 <https://doi.org/10.2135/cropsci2000.401256x>). Later, the 1A/1C translocation was found to be associated with the crown freezing tolerance and winter field survival in an RIL population derived from a cross of the non-winter-hardy ‘Fulghum’ (non-1A/1C) with winter-hardy ‘Norline’ (1A/1C) (*Crop Sci* 2006 <https://doi.org/10.2135/cropsci2005.0152>; *Crop Sci* 2007 <https://doi.org/10.2135/cropsci2006.12.0768>). These findings in oat, together with the reports from other species, support the possible importance of chromosome rearrangement in shaping local environmental adaptation and speciation. We further performed GO analyses of the genes within the large translocations observed. As mentioned above, the results show that the top-ranking biological process terms for genes within the six large translocations in tetraploid are ‘multicellular organism development’, ‘ubiquitin-dependent protein catabolic process’, and ‘oxidation-reduction process’ (Supplementary Table 15), while genes within the 1A/1C translocation were significantly enriched for ‘photosynthesis, light harvesting’ and ‘regulation of systemic acquired resistance’ (Supplementary Table 16). These processes are important for responses to abiotic and biotic stress, and thus might be involved in local environmental adaptation and (or) speciation in oat. Even so, we still agree with the reviewer that more evidences are needed to validate the functional enrichments in these regions. Therefore, we added details in the revised manuscript to make it more credible and rewrote this section to make it more moderated at this point [L246-263]:

“Chromosomal rearrangements have been implicated as one of the driving forces for shaping adaptation and speciation by affecting gene expression through position effects²⁶⁻²⁸. In oat, the 1C/1A translocation (previously designated as 7C/17A) is well-known to be associated with the division of cultivated oat into *A. sativa* L and *A. byzantina* K. Koch (sub)species²⁹ and variations in crown freezing tolerance and winter field survival^{30,31}. Functional enrichment analyses showed that ‘multicellular organism development’ (GO:0007275, 37/112, BH-Adjusted $P = 9.69E-12$), ‘ubiquitin-dependent protein catabolic process’ (GO:0006511, 51/218, BH-Adjusted $P = 4.84E-09$), and ‘oxidation-reduction process’ (GO:0055114, 373/3,273, BH-Adjusted $P = 3.00E-06$) are the top three most enriched biological processes terms for genes positioned within the six large inter-genomic translocations that occurred during tetraploidization (Supplementary Table 15). Genes positioned within the 1A/1C translocation were significantly enriched for two biological process terms

‘photosynthesis, light harvesting’ (GO:0009765, 11/139, BH-Adjusted $P = 2.92E-02$) and ‘regulation of systemic acquired resistance’ (GO:0010112, 5/23, BH-Adjusted $P = 2.92E-02$) (Supplementary Table 16). These processes are essential for responses to abiotic and biotic stress, and thus might be involved in local environmental adaption and (or) speciation in oat.”

[2. Potential issues with the analyses]

There are quite a few places in the manuscript where the analyses appear to not be appropriate for the conclusions the authors draw. For example: — [114] how many bases were callable? It looks from this stat that the entire genome seq was used as the denominator. Same goes for H0 in the next sentence. Given the large percent of the genome is repetitive, I imagine uniquely mapping reads cover a small percent of the genome.

RESPONSE: We did not filter out these multiple mapped reads before calling variants due to the concern pointed out by reviewer. This method has been widely used by other studies to evaluate the quality of genome assemblies (*Nat Genet* 2021 <https://doi.org/10.1038/s41588-021-00808-z>; *Nat Commun* 2021, <https://doi.org/10.1038/s41467-021-22920-8>; *Nat Genet* 2021, <https://doi.org/10.1038/s41588-021-00910-2>).

We also thank the reviewer for bringing ‘callable’ to our attention. Following the methods from Pan *et al.* (*Genome Biol* 2022 <https://doi.org/10.1186/s13059-021-02569-8>), the callable regions of each oat genome were identified. Briefly, the BAM files were used to generate a region file having aligned reads using GATK Callable Loci with cutoffs of minimum depth of 5, maximum depth of 300, minimum mapping quality of 20, and minimum base mapping quality of 20. The estimated heterozygosity and accuracy results based on the callable regions (Tables R4 below) and the entire genome sequence (Tables R5 below) are similar in magnitude.

Table R4 | The number of homozygous and heterozygous obtained from the callable regions, and the estimated base-level accuracy and heterozygosity.

Species	Callable Size	Callable Rate(%)	Hetero SNPs	Hetero InDels	Estimated heterozygosity	Homo SNPs	Homo InDels	Estimated error rate	Estimated accuracy
A. longiglumis	2,768,948,091	74.10	600,348	30,970	0.023%	28,967	5,037	0.001%	99.999%
A. insularis	6,038,433,804	80.31	988,863	62,963	0.017%	57,817	15,223	0.001%	99.999%
Sanfensan	8,937,643,902	83.08	583,023	35,697	0.007%	12,831	3,052	0.000%	100.000%

Table R5 | The number of homozygous and heterozygous variants obtained from the entire genomic regions without filtering out the multiple mapped reads, and the estimated base-level accuracy and heterozygosity based on entire genome sequence.

Species	Genome Size	Mapping rate(%)	Hetero SNPs	Hetero InDels	Estimated heterozygosity	Homo SNPs	Homo InDels	Estimated error rate	Estimated accuracy
A. longiglumis	3,736,548,545	99.75	1,408,256	60,259	0.039%	101,294	24,984	0.003%	99.997%
A. insularis	7,519,018,440	99.69	1,851,651	64,623	0.025%	146,513	34,835	0.002%	99.998%
Sanfensan	10,757,433,345	99.77	1,093,198	66,019	0.011%	98,885	19,444	0.001%	99.999%

— [148] why are you using single copy orthologs (I think you mean orthogroups). These will clearly be biased to genes that have sub/non functionalized in your tetraploids, potentially completely biasing all downstream analyses. Especially since your run is with the old orthofinder, which defines orthogroups as those genes that have a single common ancestor among all sampled species. Either drop your polyploids from this run, or use ploidy aware single copy orthologs.

RESPONSE: Sorry for the unclear description on how to deal with the polyploids when searching for the single copy genes. As reviewer suggested, we did drop polyploids from the run of orthofinder. The polyploid species included in this study were divided into different subgenomes (two subgenomes for tetraploid and three subgenomes for hexaploid species), each subgenome was deemed as independent taxa to search for single copy genes. We have now revised description about the samples related to orthologs identification to avoid misunderstanding [L151-156, L749-750; Supplementary Note L489-490]:

“Gene family analysis across 23 subgenomes from 16 species belonging to the BOP clade identified 2,237 single-copy orthologs (Supplementary Tables 7-9). Phylogenomic analyses (Fig. 2a) revealed that the divergence between Aveneae and Triticeae took place after the speciation of Oryzoideae, with the approximate times for the two events being 28.2 and 47.9 million years ago (mya), respectively.”

“Using the gene families identified by the OrthoFinder program, 2,237 one-to-one orthologous gene sets were identified for the 23 subgenomes of 16 grass species.”

“In total, 2,237 single copy orthologous gene clusters shared by all 23 subgenomes from 16 species were identified.”

— [266] what is going on here? An R of 0.08 is so significant ... I think you need to contextualize that this is just driven by massive n and very little signal. Further, how would one do anything with such small effect sizes? Some interpretation is important.

RESPONSE: We thank reviewer for pointing this out. We agree with the reviewer that the significance test was largely affected by the massive n. In the revised manuscript we now use the method described for strawberry (*Nat Genet* 2019 <https://doi.org/10.1038/s41588-019-0356-4>) to evaluate the relationship between TE density and gene expression level. Briefly, we equally divided the 12,225 homoeologous triads into four groups based on their associated TE densities (the equal division was to eliminate the size effect on the significance test), and then compared their gene expressions values using Student’s *t*-tests. We found that genes with relatively higher

20TE densities tend to have lower expression levels (Fig. R5 below, also Revised Extended data Fig. 10c). We have now revised the description as suggested [L308-317]:

“Fourth, we found that the C subgenome contained more transposable elements (TEs) (A vs C vs D, 2.88 Gb vs 3.61 Gb vs 2.76 Gb) (Supplementary Table 6) and showed a higher overall TE density near genes than did the A and D subgenomes (C vs A, 0.454 vs 0.438, $P = 4.595E-09$; C vs D, 0.454 vs 0.437, $P = 4.595E-09$, Student’s t -test, $df = 24448$) (Extended Data Fig. 10b). We further found that genes with relatively higher TE densities near genes tend to have lower expression levels (Extended Data Fig. 10c, pairwise Student’s t -test). This observation is consistent with the hypothesis that subgenome gene expression dominance is influenced by TE-density differences between subgenomes as observed in other allopolyploids^{8,32}.”

Fig. R5 | The relationship between TE density and gene expression. A pairwise test (Student’s t -test, $P < 0.05$) was used to determine significance (Revised Extended data Fig. 10c).

Statistics are generally not reported or incomplete. I mentioned this clearly in my previous review, the authors state that they fixed it, but it is far from fixed. For example (again not an exhaustive list):

[218] here and elsewhere, when a comparison is made, statistical support needs to be provided.

RESPONSE: We have now explicated the nature of the statistical support in the revised manuscript [L234-238]:

“Consistent with the inference that the A and D subgenomes of oats are closely related, the synteny between them (54.40% genes in collinear blocks) is more extensive ($P < 2.2E-16$, Fisher’s exact test) than that between the C and A/D subgenomes in

hexaploid (43.59% in C and A, 41.43% in C and D) and tetraploid (38.81% in C and D) (Extended Data Fig. 5)”

[247] no stats again in this whole paragraph

RESPONSE: We have now added statistics throughout this whole paragraph and the related figure captions [L277-317].

[259] what test, stat and df? Pvalue alone is not sufficient

RESPONSE: We have now added these details as suggested [L295-300]:

“Third, examination of the orthologs expression patterns showed that the number of preferentially expressed genes in the C subgenome was significantly lower than that in the A (up-regulated genes in A vs C, 9375 vs 7541, $P=6.04E-08$, Wilcoxon rank-sum test) and D (D vs C, 9359 vs 8139, $P=8.32E-07$, Wilcoxon rank-sum test) subgenomes (Fig. 4f-g, Supplementary Table 20).”

In addition, we have included the specific inferential statistical tests used to each of the relevant figure captions (Fig. 4, Fig. 6, and Extended Data Figs. 8-10)

- - - [3. Phylogenomic analyses] - - -
In the previous draft I reviewed, the authors conducted an across-flowering plant analyses, and it wasn't clear to me why they chose so many species when the proximate goal of all methods in this paper is to understand *Avena* evolution. Although the authors have now restricted the analyses to grasses, it remains unclear to me why so many non-avena species are here. No justification is presented and no analyses are explicitly discussed outside of the *Avena* clade. So, I'll repeat my previous comment with some minor changes: “The purpose of the analysis outside of *Avena* (and one or two outgroups) was not obvious to me. As a reader, I had trouble grasping how divergence time between (for example) [*Sorghum* and *Maize*] integrated with the primary goals of this paper, namely the origin and evolution of the oat genome. How do your assemblies in *Avena* have any bearing on phylogenetic analysis of other clades? These comments apply to the [poaceae] gene PAV (presence / absence variations) analysis, phylogenetic analyses [143-152] and several other sections.”

RESPONSE: We now appreciate your larger point here. We have accordingly reduced the number of out groups in the analysis and focused specifically within Pooidae for our related claims (Fig. R6 below, also Revised Main text Fig. 2a-b). We have now explicitly explained our aim of positioning oats amongst the Poaceae and have added relevant discussion [L144-162]:

“The Poaceae family consists of many agronomically important species, commonly known as cereals, that are classified into three subfamilies: Oryzoideae (rice), Panicoideae (maize, sorghum), and Pooideae (Triticeae: wheat, barley and rye; Aveneae: oat). Among them, Oryzoideae and Pooideae belong to BOP clad (Bambusoideae, Oryzoideae, and Pooideae), which is one of the two primary groups in Poaceae family. To further clarify the evolutionary position of oat among cereal crops, we used our oat reference genome assemblies to perform the first phylogenomic analysis to include oat. Gene family analysis across 23 subgenomes from 16 species belonging to the BOP clade identified 2,237 single-copy orthologs (Supplementary Tables 7-9). Phylogenomic analyses (Fig. 2a) revealed that the divergence between Aveneae and Triticeae took place after the speciation of Oryzoideae, with the approximate times for the two events being 28.2 and 47.9 million years ago (mya), respectively. The tribes Aveneae and Lolieae were indicated to be more closely related than Triticeae. The diversification of oat species occurred ~8.7 mya, which is earlier than wheat species (~6.6 mya) and falls within the previously estimated speciation time of *Avena* diploids (5.4-12.9 mya)¹⁸. We compared the gene families among Aveneae, Triticeae, and Lolieae, and found 6425 common gene families in these three tribes and there are 1608 gene families specific to Aveneae (Fig. 2b).”

Fig. R6 | Phylogenomic relationships of cereal crops. **a**, Phylogeny and time scale of 23 subgenomes from 16 cereal crops and related grass species, including panicle images of the 11 sampled cereal crops. The number in the upper left corner of each panicle image corresponds to the number following the species name in the phylogenetic tree. **b**, Venn diagram showing the numbers of shared and unique gene families in the tribes of Aveneae, Lolieae, and Triticeae (Revised Main text Fig. 2a-b).

I still don't understand why you used an ancestral karyotype for any analyses. Why not reconstruct it for *Avena* and have that as an added value for this paper. I disagree with this statement: "The use of AGK allowed us to trace the history of oat chromosomal structural rearrangements more easily under the framework of phylogeny." This would add error to the analysis, since you could just use barley as the outgroup to phase the three oats. Or maybe this is a problem with communication (see above), and the ancestral oat genome was used, not the ancestral grass genome ... for example [159] is the AGK used here the one from the split between *avena* and *loium* (or maybe barley)? It appears this is not the case and that it is the ancestral karyotype to all grasses, which would add 30M years of error to this analysis. Why did you choose to use this ancestral state that far predated the origin of *avena* when the only chr evolution discussed is within *Avena*?

RESPONSE: We thank the reviewer for drawing attention to this point. We now understand the problem here. In the revised manuscript, we have now concentrated on the BOP clade. Rice was identified as the most slowly evolving species and has 12 chromosomes, most of which closely resemble the post- ρ AGK (*Nat Genet* 2017 <https://doi.org/10.1038/ng.3813>). Based on the phylogenomic analysis, Triticeae (wheat, barley and rye) and Aveneae (oat) clustered together with Oryzoideae (rice) as the outgroup. We therefore used the rice genome as an ancestral reference to investigate karyotype origins, and used the barley genome as closely reference as suggested to compare chromosome evolution with another allohexaploid cereal species wheat (Fig. R7 below, also Revised Main text Fig. 2c-d).

Fig. R7 | *Avena* chromosome evolution based on the rice and barely genomes. c, Probable chromosome evolutionary scenario of oat and wheat species. The subgenome chromosomes (1–7) are presented with a color code to show different segments from the 12 chromosomes of rice (Os1–Os12), which can be used as the representative of the ancestral grass chromosomes (AGK1–AGK12). The abbreviations in the top of the subgenomes are accordance with the abbreviations indicated in the brackets of panel d. **d**, chromosome evolution based on barely genomes and chromosomal synteny between the three subgenomes of oat and wheat (Revised Main text Fig. 2c-d).

Also in my previous review I brought up concerns with the synteny analysis. These still remain, perhaps more so. The authors compared “genomes based on predicted protein sequences using McScan with default parameters.” McScanX with default parameters is designed to look for ancient collinearity and will recover the Rho and other paralog blocks. It appears the authors dealt with this problem by first running the defaults, then culling block to the largest. If this is the case, I am not surprised that there are lots of gaps and overlaps. This is not a good approach and the revised methods do not assuage my previous concerns - look carefully at your block breakpoints ... do they look reasonable? My guess is no, considering the tiny broken blocks in the McScanX plot and table R1’s highly incomplete comparisons.

RESPONSE: We are deeply thankful to the reviewer for raising this point as we realized that our original approach requires adjustment. Refer to the synteny analysis of wild emmer (*Science* 2017 <https://doi.org/10.1126/science.aan0032>) and wheat (*Science* 2018

25<https://doi.org/10.1126/science.aar7191>), we have now identified gene homoeologs using a hierarchical approach to obtain for each gene the best bi-directional BLAST hit, and the synteny was analyzed using the McScanX software on the homoeologous gene pairs. The regional bias was avoided by using a relaxed block size requirement (≥ 3 genes per block). Using this synteny analysis approach, we re-performed a synteny analysis of published wheat genomes (*Science* 2018 <https://doi.org/10.1126/science.aar7191>). The similarity of our McScanX plot for wheat (Fig. R8a) to the plot presented by the International Wheat Genome Sequencing Consortium (*Science* 2018 <https://doi.org/10.1126/science.aar7191>) (Fig. R8b) supports that our synteny analysis is informative.

Fig. R8 | Conserved synteny landscape of the 21 wheat chromosomes. a, Connecting lines in the center of the diagram highlight homoeologous relationships of chromosomes and translocated regions, colors assigned to different homologous groups. b, Connecting lines in the center of the diagram highlight homoeologous relationships of chromosomes (blue lines) and translocated regions (green lines) (*Science* 2018 <https://doi.org/10.1126/science.aar7191>).

We have now updated the results of synteny analysis in our revised manuscript. In order to display the results more intuitively to avoid misleading, we highlighted the synteny between each hexaploid chromosome and its corresponding ancestor chromosome in the Revised Main text Fig.1 (Fig. R9 below) to reveal relationships of the hexaploid A, C, and D subgenomes with its corresponding progenitor genomes (A genome of diploid *A. longiglumis*, C and D subgenomes of tetraploid *A. insularis*). And the syntenic relationship of inter- and intra- subgenomes within polyploids are showed in Revised Extended Data Fig. 5 (Fig. R10 below).

Fig. R9 | Circos display of the genomic features of the assembled diploid *A. longiglumis*, tetraploid *A. insularis*, and hexaploid *A. sativa ssp. nuda* cv. *sanfensan*. **a**, The inferred centromere positions of the A and D genome chromosomes. **b**, The distribution of the C genome-specific repeat Am1 along each chromosome. The Am1-rich regions on chromosomes 1A, 2D, 3D, 4D, and 5D of ‘Sanfensan’ are C genome introgressions. **c**, The distribution of the A genome-specific repeat As120a along each chromosome. **d**, k-mer frequencies. **e**, Tandem repeat (TR) density (MaxPeriod ≤ 500 bp). **f**, Long terminal repeat (LTR) retrotransposon density. **g**, Gene density. **h**, Chromosome names and sizes. The innermost layer shows synteny of hexaploid and its ancestor species, with colored upper-layer links representing syntenic blocks of each hexaploid chromosome and its

27ancestor chromosome, and the gray low-layer shows chromosome rearrangements during hexaploidization (Revised Main text Fig.1).

Fig. R10 | Structural and conserved synteny landscape of the ‘Sanfensan’ (a) and *A. insularis*

(b) genomes. Homoeologous gene pairs in syntenic blocks (>20 genes) are linked. The four rings depict the 31-mer distribution along each chromosome (1), the density of tandem repeats (TRs, motif length ≤ 500 bp) (2), the density of long terminal repeats (LTRs) (3), and the density of protein-coding genes (4) (Revised Extended Data Fig. 5).

It is obvious that the synteny among homoeologous chromosomes in the hexaploid oat is lower and more fragmented than in wheat, which consistent with the percentages of homoeologs within syntenic blocks (56.71% in oat vs 71.57% in wheat), and the percentages of wheat genes within syntenic blocks from this study are in accordance with the related result from the International Wheat Genome Sequencing Consortium which indicated that 72% of the homeologs were organized in collinear blocks (*Science* 2018 <https://doi.org/10.1126/science.aar7191>). In addition, we investigate the percentages of homoeologs within syntenic blocks between different subgenomes of oat and wheat. The results showed that the collinearity of wheat subgenomes (59.86% in A and B, 64.08% in A and D, 63.52% in B and D) is higher overall than in oat subgenomes, and the collinearity between A and D (54.40% genes in collinear blocks) is more extensive ($P < 2.2E-16$, Fisher's exact test) than that between the C and A/D subgenomes in hexaploid (43.59% in C and A, 41.43% in C and D) and tetraploid (38.81% in C and D) oats. These results are consistent with the MCScanX plot which showed that all the C subgenome have larger sized gaps compared to A/D subgenome of polyploid oat, suggesting the method used is appropriate.

- - - [4. Minor/specific comments] - - -
 — [106] what does transcript % mean?

RESPONSE: Sorry for the unclear statement. Regarding the data in old line 106, the “transcript %” here means the genome assembly covered 92% of the transcripts generated by Iso-seq, indicating coverage of most of the gene regions. We have now added more specific statements in the revised manuscript [L101-104]:

“The quality of the ‘Sanfensan’ genome assembly was supported by assessments including NG plot, long terminal repeat (LTR) Assembly Index (LAI, 18.34), BUSCO (99.44%) and the coverage of full-length transcripts by the assembled genome (92.00%) (Extended Data Fig. 2)”

— [255] what progenitors? I’m not following this logic

RESPONSE: We thank you for bringing our attention to this. We have clarified this point as “A- and C-genome diploid progenitors” in the revised manuscript [L293-294].

— [291] what is the scale of LD in this pop? This is a very wide interval for standard GWAS. Either you have truly unparalleled LD or your model is misspecified, perhaps with uncontrolled population structure or non normal phenotypes. The methods appear sound, so not sure what's up here

RESPONSE: We thank the reviewer for this excellent guidance. The population structure inferred from PCA and NJ tree (Fig. R11b-c below, also Revised Main text Fig. 6b-c) showed with relatively weak population structure for the 659 oat lines. Genome-wide LD decay rate of this population was estimated at ~2.29 Mb (Fig. R11d below, also Revised Main text Fig. 6d). The long-range LD observed in oat was similar to that in other self-fertilizing species such as wheat (*Euphytica* 2016 <https://doi.org/10.1007/s10681-016-1750-y>; *PLoS One* 2021 <https://doi.org/10.1371/journal.pone.0246015>) and barley (*BMC Plant Biol* <https://doi.org/10.1186/1471-2229-12-16>; *Mol Breed* 2017 <https://doi.org/10.1007/s11032-017-0626-8>).

For the GWAS analysis, we used a mixed-linear model which incorporated population structure and kinship matrix as cofactors to reduce putative false positives. In this step, the top five principal components were used to build up the P matrix for population structure correction. The quantile-quantile plot showed that the observed P value corresponded to the expected values, suggesting the model used is appropriate (Fig. R11e below, also Revised main text Fig.6e). The GWAS scan detected a strong peak on chromosome 4D to be highly associated with the hullless grain trait. Most (16/24) of the significantly associated makers on 4D were distributed in a region between 438-450 Mb, thereby, we just roughly defined this region in our original manuscript. To avoid misleading, we have now displayed the physical positions of the flanking markers to the *NI* locus previously reported (*Euphytica* 2017 <https://doi.org/10.1007/s10681-017-1836-1>) and the mostly associated marker detected in the present study (Fig. R11f below, also Revised Main text Fig. 6f)

Fig. R11 | Population structure of 659 oat lines and the pattern of LD. **b**, Scatter plots of the first principal component (PC1) versus PC2 (left) and PC1 versus PC3 (right). **c**, Neighbor-joining tree constructed based on the Euclidean distance metrics between taxa. **d**, LD decay across the whole genome. The r^2 values of LD were plotted against the physical distance in Mb. LD dropped to 0.2 at 2.29 Mb across the whole genome (Revised Main text Fig. 6b-d). **e**, Quantile-quantile plot of the MLM model for hullless grain trait. **f**, Manhattan plot for candidate region association mapping of *NI*. The zoomed-in region displays the physical positions of the markers GMI_ES22_c7478_431 and GMI_ES14_c19259_657 flanking the *NI* locus⁴², and the mostly associated markers (Revised Main text Fig. 6b-f).

We have now added the information on population structure and LD in the GWAS population to the revised manuscript [L337-345] as follows:

“We first infer the population structure and LD pattern of the oat collection. Both principal component analysis (PCA) (Fig. 6b) and Neighbor-joining (NJ) tree (Fig. 6c) revealed weak population structure, which is consistent with previous studies³⁵⁻³⁷. Most of the hullless landraces were tightly clustered reflecting domestication

bottleneck for hulless oat³⁴. The LD decay rate was measured as the physical distance at which the average pairwise r^2 dropped to 0.2. Genome-wide LD decay rate of this population was estimated at 2.29 Mb (Fig. 6d), The long-range LD observed in oat is similar to that in other self-fertilizing species such as wheat^{38,39} and barley^{40,41}.”

— [292-293] specific language again. What was examined and why exactly was this gene chosen? The reader needs enough detail to be able to recreate the analysis that resulted in this being the strongest candidate

RESPONSE: Following the guidance of the reviewer, we now provide more details on candidate gene prediction in the revised manuscript. In the present study, genome-wide association scan detected a strong peak on chromosome 4D associated with the hulless grain trait, and the most highly associated marker was located at 450.96 Mb (Fig. R12b, also Revised Main text Fig. 6f). Many studies have suggested the hulless grain trait is controlled by a major gene termed *Naked1 (NI)* (*Euphytica* 1976 <https://doi.org/10.1007/BF00041542>; *Can J Plant Sci* 1994 <https://doi.org/10.4141/cjps94-090>; *Euphytica* 2017 <https://doi.org/10.1007/s10681-017-1836-1>), which was previously mapped to an interval of 0.7 cM flanked by markers GMI_ES14_c19259_657 and GMI_ES22_c7478_431 (*Euphytica* 2017 <https://doi.org/10.1007/s10681-017-1836-1>). The physical locations of these two markers on the ‘Sanfensan’ chromosomes were determined by BLASTn analysis. We found the GMI_ES14_c19259_657 and GMI_ES22_c7478_431 are highly similar to genomic sequences located at 451.22 Mb and 449.02 Mb on chromosome 4D, respectively. These results suggested the hulless grain peak detected in our study is collocated with the *NI* locus.

We then searched the region around the peak for plausible candidate genes based on their predicted functions, and paid particular attention to genes within the genomic region covered by flanking markers mentioned above. A gene (gene ID: *A.satnudSFS4D01G000045*) annotated as a receptor-like kinase came across. A homologous gene *AtVRLK1* in *Arabidopsis* functions in thickening the secondary cell wall (*Plant Physiol* 2018 <https://doi.org/10.1104/pp.17.01279>), while a mutant of the homologous gene *mis2* in rice displayed an open hulled spikelet (*Rice* 2020 <https://doi.org/10.1186/s12284-020-0368-9>). We found this gene is very close (~30 kb distant) to the most highly associated marker and thereby considered it as the major candidate gene under *NI* locus.

We next compared the genomic sequences of this gene in hulled and hulless oats, and found 25 single nucleotide changes in the coding regions with one SNP in exon1 predicted to cause an amino acid change (Fig. R12d below, also Revised Main text Fig. 6g). This SNP was converted to a KASP marker and validated in 286 oat lines randomly selected from the oat germplasm collection subjected to GBS sequencing: the hulless oats could be clearly separate from the hulled oats using

this KASP marker (Fig. R12e below, also Revised Main text Fig. 6h), confirming the close association between *A.satnudSFS4D01G000045* and the hullless grain trait.

Further, we compared transcriptome data from RNA samples equally mixed from seven tissue/conditions between hulled and hullless oats, and found that *A.satnudSFS4D01G000045* was differentially expressed between hulled and hullless oats (Fig. R12f below, also Revised main text Fig. 6i). We subsequently examined the expression levels of this gene in panicles of ‘Sanfensan’ (hullless) and ‘Ogle’ (hulled) during different developmental stages including the booting (Zadok’s 45), heading (Zadok’s 50 and 58), and grain dough (Zadok’s 83) stages by RNA-seq. The results showed this gene was highly expressed in panicles of “Sanfensan” (hullless) during all four developmental stages, but was expressed at very low levels in panicles of “Ogle” (hulled) (Fig. R12g below, also Revised main text Fig. 6j). All of these results suggest that *A.satnudSFS4D01G000045* is a promising plausible candidate gene for control of the hullless grain trait in oat. We have re-written this paragraph in the revised manuscript [345-364] as follows:

“An association scan detected a strong peak on the end of chromosome 4D, which collocated with the previously reported *NI* locus^{34,42} (Fig. 6e-f). We searched the vicinity of the peak for plausible candidate genes with particular attention to genes within the genomic region covered by markers flanking the *NI* locus⁴² (Fig. 6f), and came across a gene (*A.satnudSFS4D01G000045*) annotated as a receptor-like kinase (RLK) which is about 30 kb distant from the most highly associated marker. A homologous gene *AtVRLK1* in Arabidopsis was found to be involved in secondary cell wall thickening⁴³, and the mutant of the homologous gene *mis2* in rice displayed an open hulled spikelet⁴⁴. Comparison the coding sequences of *A.satnudSFS4D01G000045* identified a SNP in first exon predicted to cause amino acid changes (Fig. 6g). The association of this SNP with the hullless trait was validated by the development of KASP (Kompetitive Allele-Specific PCR) markers (Fig. 6h). By comparing transcriptome data between 10 hulled and 12 hullless oats, we found *A.satnudSFS4D01G000045* is differentially expressed with hullless oats having higher expression levels ($P < 0.01$, Student’s *t*-test) (Fig. 6i). Further examination of the expression patterns in the panicles at different developmental stages revealed this gene is highly expressed during panicle development in ‘Sanfensan’ (hullless), but is expressed at very low levels in panicles of ‘Ogle’ (hulled) (Fig. 6j, Student’s *t*-test). These results indicated that *A.satnudSFS4D01G000045* may be a promising plausible candidate gene that controls the hulled/hullless trait in oat.”

Fig. R12 | Genome-wide scan for the hulless grain trait. **a**, The spikelets and kernels of hulled (left) and hulless (right) oats. **b**, Manhattan plot for candidate region association mapping of *NI*. The zoomed-in region displays the physical positions of the markers GMI_ES22_c7478_431 and GMI_ES14_c19259_657 flanking the *NI* locus⁴², and the mostly associated markers. **c**, Quantile-quantile plot of the MLM model for hulless grain. **d**, Gene structure of the candidate gene *A.satnudSFS4D01G000045*. Green boxes are exons, and lines between the boxes are introns. The location of a SNP that resulted in an amino acid change is shown in the diamond shaped box. **e**, The association between *A.satnudSFS4D01G000045* and the hulless trait was validated using a KASP marker derived from the SNP. **f**, The expression level of *A.satnudSFS4D01G000045* in RNA samples equally mixed from seven tissue/conditions of 10 hulled and 12 hulless oat lines. The central line for each box plot indicates the median. The top and bottom edges of the box indicate the first and third quartiles, and the whiskers extend 1.5 times the interquartile range beyond the edges of the box. **g**, Comparison of expression levels of *A.satnudSFS4D01G000045* in panicles at different developmental stages between ‘Sanfensan’ (hulless) and ‘Ogle’ (hulled). The data are presented as mean±s.d. (n=3 biological replicates). S1, S2, S3, and S4 represent panicles

34at the booting (Zadok's 45), heading (Zadok's 50 and 58), and grain dough (Zadok's 83) stages, respectively. Student's *t*-test was used to generate the *P* values in **f** and **g** (*, $P < 0.05$, **, $P < 0.01$) (Revised Main text Fig. 6).

Finally, let us again sincerely thank the reviewer for the time and efforts to give us the helpful guidance about how to improve our study and manuscript.

Reviewer #3:
Remarks to the Author:
The authors have addressed all my requests for additional analysis and clarifications. They have added a substantial number of additional analysis and data.

In particular, novel analysis of important agronomic traits is included, adding value to the genomic analysis. Furthermore, a careful comparative analysis to existing data is presented, adding further value.

The manuscript will be of high value for the genomics community.

RESPONSE: Thank you very much for approving our revised manuscript which has been further improved with your help. We appreciate your great contribution to our revision.

Decision Letter, first revision:

Our ref: NG-A59600R

12th May 2022

Dear Dr. Peng,

Thank you for submitting your revised manuscript "Reference genome assemblies reveal the origin and evolution of allohexaploid oat" (NG-A59600R). It has now been seen by the original referees and their comments are below. The reviewers find that the paper has improved in revision, and therefore we'll be happy in principle to publish it in Nature Genetics, pending minor revisions to comply with our editorial and formatting guidelines.

We are now performing detailed checks on your paper and will send you a checklist detailing our editorial and formatting requirements soon. Please do not upload the final materials and make any

35revisions until you receive this additional information from us.

Sincerely,
Wei

Wei Li, PhD
Senior Editor
Nature Genetics
New York, NY 10004, USA
www.nature.com/ng

Reviewer #1 (Remarks to the Author):

The authors have adequately addressed my concerns.

Final Decision Letter:

In reply please quote: NG-A59600R1 Peng

8th Jun 2022

Dear Dr. Peng,

I am delighted to say that your manuscript "Reference genome assemblies reveal the origin and evolution of allohexaploid oat" has been accepted for publication in an upcoming issue of Nature Genetics.

Due to the importance of these deadlines, we ask that you please let us know now whether you will be

36difficult to contact over the next month. If this is the case, we ask you provide us with the contact information (email, phone and fax) of someone who will be able to check the proofs on your behalf, and who will be available to address any last-minute problems.

Your paper will be published online after we receive your corrections and will appear in print in the next available issue. You can find out your date of online publication by contacting the Nature Press Office (press@nature.com) after sending your e-proof corrections. Now is the time to inform your Public Relations or Press Office about your paper, as they might be interested in promoting its publication. This will allow them time to prepare an accurate and satisfactory press release. Include your manuscript tracking number (NG-A59600R1) and the name of the journal, which they will need when they contact our Press Office.

Please note that *Nature Genetics* is a Transformative Journal (TJ). Authors may publish their research with us through the traditional subscription access route or make their paper immediately open access through payment of an article-processing charge (APC). Authors will not be required to make a final decision about access to their article until it has been accepted. [Find out more about Transformative Journals](https://www.springernature.com/gp/open-research/transformative-journals)

Authors may need to take specific actions to achieve [compliance with funder and institutional open access mandates](https://www.springernature.com/gp/open-research/funding/policy-compliance-faqs). If your research is supported by a funder that requires immediate open access (e.g. according to [Plan S principles](https://www.springernature.com/gp/open-research/plan-s-compliance)) then you should select the gold OA route, and we will direct you to the compliant route where possible. For authors selecting the subscription publication route, the journal's standard licensing terms will need to be accepted, including [self-archiving-and-license-to-publish](https://www.nature.com/nature-portfolio/editorial-policies/self-archiving-and-license-to-publish). Those licensing terms will supersede any other terms that the author or any third party may assert apply to any version of the manuscript.

Please note that Nature Portfolio offers an immediate open access option only for papers that were first submitted after 1 January, 2021.

If you have any questions about our publishing options, costs, Open Access requirements, or our legal

forms, please contact ASJournals@springernature.com

If you have not already done so, we invite you to upload the step-by-step protocols used in this manuscript to the Protocols Exchange, part of our on-line web resource, natureprotocols.com. If you complete the upload by the time you receive your manuscript proofs, we can insert links in your article that lead directly to the protocol details. Your protocol will be made freely available upon publication of your paper. By participating in natureprotocols.com, you are enabling researchers to more readily reproduce or adapt the methodology you use. [Natureprotocols.com](http://natureprotocols.com) is fully searchable, providing your protocols and paper with increased utility and visibility. Please submit your protocol to <https://protocolexchange.researchsquare.com/>. After entering your nature.com username and password you will need to enter your manuscript number (NG-A59600R1). Further information can be found at <https://www.nature.com/nature-portfolio/editorial-policies/reporting-standards#protocols>

Sincerely,
Wei

Wei Li, PhD
Senior Editor
Nature Genetics
New York, NY 10004, USA
www.nature.com/ng